# Hierarchical Subspaces of Policies for Continual Offline Reinforcement Learning

## Abstract

We consider a Continual Reinforcement Learning setup, where a learning agent must continuously adapt to new tasks while retaining previously acquired skill sets, with a focus on the challenge of avoiding forgetting past gathered knowledge and ensuring scalability with the growing number of tasks. Such issues prevail in autonomous robotics and video game simulations, notably for navigation tasks prone to topological or kinematic changes. To address these issues, we introduce `HiSPO`, a novel hierarchical framework designed specifically for continual learning in navigation settings from offline data. Our method leverages distinct policy subspaces of neural networks to enable flexible and efficient adaptation to new tasks while preserving existing knowledge. We demonstrate, through a careful experimental study, the effectiveness of our method in both classical MuJoCo maze environments and complex video game-like navigation simulations, showcasing competitive performances and satisfying adaptability with respect to classical continual learning metrics, in particular regarding the memory usage and efficiency.
https://sites.google.com/view/hierarchical-subspaces-crl/

## 1 Introduction

Humans continuously acquire new skills and knowledge, adapting to an ever-changing world while retaining what they have previously learned. Designing systems capable of replicating this lifelong learning ability is a key challenge in Continual Reinforcement Learning (CRL) (Khetarpal et al., 2022), as traditional Reinforcement Learning (RL) (Sutton & Barto, 2018) may struggle with adaptive and cumulative learning. In CRL, a learning agent must sequentially solve tasks, requiring to master new skills without degrading the knowledge gained from previous tasks.

Within this framework we focus Goal-Conditioned RL (GCRL) (Ding et al., 2019; Liu et al., 2022), involving learning policies that can be conditioned to reach specific goal states, making it relevant for real-world applications in robotics and video games where navigation is crucial. The *offline* setting (Levine et al., 2020; Prudencio et al., 2023), which relies on pre-collected data, is particularly appealing when data collection is expensive, risky, or impractical. However, alone, this setting is not sufficient in the context of changing environments: agents need to continuously adapt to new tasks without forgetting the previous ones, while maintaining scalability as the number of tasks increases (Graffieti et al., 2022; Shaheen et al., 2022).

Various CRL methods have been proposed to tackle these challenges : some use replay buffer or generative models to replicate past tasks (Rolnick et al., 2019) ; others involve architectural revisions to mitigate forgetting (Rusu et al., 2016) ; and some use regularization techniques to improve scalability (Kirkpatrick et al., 2017; Kumar et al., 2023). Nevertheless, these approaches face limitations : Replay-based methods can be impractical due to limited data storage and privacy constraints, particularly in industrial applications where long term data retention may be costly. Regularization techniques struggle with highly diverse changes, and architecture modifications, such as expanding neural network structures, can become memory-intensive thus limiting scalability. While entirely addressing all these limitations is challenging, Continual Subspace of Policies (CSP) (Gaya et al., 2023) stands out as an interesting balance between flexibility and efficiency, using subspaces of neural networks (Wortsman et al., 2021; Gaya et al., 2022), which help adapting without forgetting previous acquired skills. However, it is primarily an online method and remains untested in offline settings where it may face new challenges.

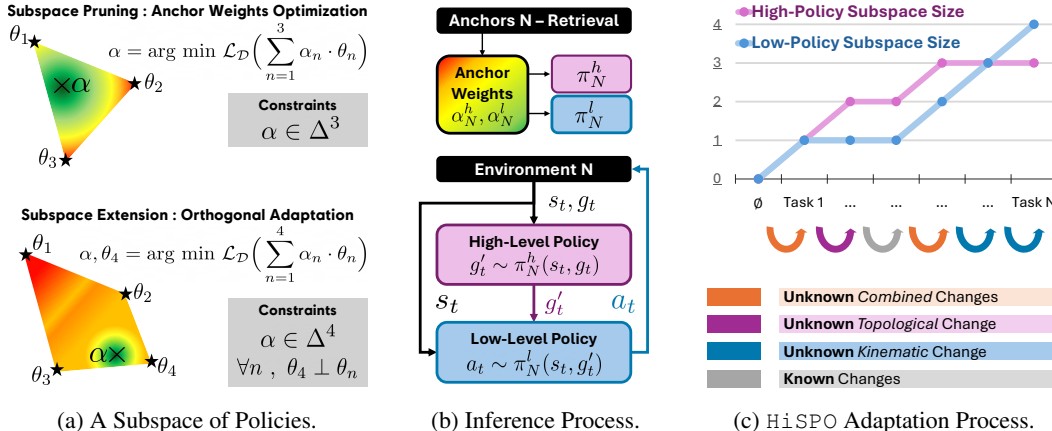

(a) A Subspace of Policies.  (b) Inference Process.  (c) `HiSPO` Adaptation Process.

Figure 1: **Hierarchical Subspaces of Policies (`HiSPO`) :** (a) *Pruning* and *Extension* mechanisms. *Pruning* involves optimizing anchor weights $\alpha$ within a defined simplex, allowing efficient exploration of the existing subspace. *Extending* introduces new anchors to expand the subspace, facilitating the adaptation to new tasks while keeping a compact representation of parameters. **(b) The inference pipeline leveraging learned anchors.** The high-level policy generates sub-goals, which the low-level policy follows by producing adequate actions. **(c) Memory-efficient adaptation process.** High-level and Low-level policy subspaces expand as new tasks introduce unknown changes, either *Topological* (affecting path planning) or *Kinematic* (affecting local actions).

In this article, we propose the **Hierarchical Subspaces of Policies (`HiSPO`)** framework, a practical offline adaptation of Continual Subspace of Policies (CSP) for hierarchical architectures, which is particularly well suited for goal-conditioned tasks. `HiSPO` novelty notably dwells on growing separate parameter subspaces, for a high-level path-planning policy and a low-level path-following policy, depending on the task stream (see *Figure 1*).

While *Section 2* reviews the relevant and related literature, *Section 3* present the theoretical background that help contextualizing our research. In *Section 4*, we define and detail our proposed approach. *Sections 5.1* and *5.2* present our experimental methodology, comparing our approach in both novel video-game-like settings with human-authored datasets and classical goal-conditioned environments. Finally, *Sections 5.3* to *5.5* present our experimental results. Our main contributions are :

- `HiSPO`, a novel hierarchical framework for Continual Offline Reinforcement Learning, leveraging two distinct subspaces of policies for scalable low-memory adaptation regarding navigation tasks.

- A large panel of Goal-Conditioned navigation tasks with datasets, encompassing both robotics and video games scenarios with human-authored datasets. We hope this new light-weighted open-source benchmark will provide a comprehensive testing ground for future research.

- A comprehensive experimental evaluation of `HiSPO` and state-of-the-art CRL algorithms using our proposed benchmark. Our results show competitive scalability and adaptability of our method, showcasing its ability to handle diverse and complex tasks across various metrics.

## 2 RELATED WORK

We review methods across learning paradigms to clarify how our work uniquely addresses the challenges of Offline Continual Reinforcement Learning (CRL) in goal-conditioned tasks.

**Transfer, Multitask, and Meta-Learning** (Zhu et al., 2023; Vithayathil Varghese & Mahmoud, 2020; Beck et al., 2023) leverage shared knowledge to improve learning efficiency. However, they typically require simultaneous access to all tasks and do not naturally handle the sequential, evolving nature of CRL. In contrast, **Continual Reinforcement Learning (CRL)** targets sequential task learning while preventing catastrophic forgetting (Díaz-Rodríguez et al., 2018; Khetarpal et al., 2022). Approaches include : **Replay-Based** methods, which mitigate forgetting by storing and replaying past experiences (Rolnick et al., 2019), but face storage and privacy issues ; **Regularization** techniques (e.g., EWC (Kirkpatrick et al., 2017), L2 (Kumar et al., 2023)) that constrain parameter updates, though they can struggle with highly diverse tasks ; **Architectural** strategies (e.g., Progressive Neural Networks (Rusu et al., 2016)) that isolate task-specific parameters but may not scale well.

Most work in CRL has focused on the **Online Setting** (Wang et al., 2024), while **Offline CRL** — learning from fixed datasets (Isele & Cosgun, 2018; Liu et al., 2024) — remains less explored, especially for goal-conditioned tasks. Standardized benchmarks for this setting are lacking.

Architectural approaches like **Continual Subspace of Policies (CSP)** (Gaya et al., 2022; 2023) and **Low-Rank Adaptation (LoRA)** (Hu et al., 2021) have shown promise in both online and offline scenarios. However, their simultaneous application to offline CRL has been limited.

**Hierarchical Policies (HP)** are effective for goal-conditioned and multi-step planning tasks (Gupta et al., 2019; Park et al., 2023), decomposing decision-making into manageable levels. While many HP methods rely on complex models or focus on meta- and multitask learning (Pan et al., 2024; Shu et al., 2018; Chua et al., 2023), our approach integrates lightweight HP into an offline CRL framework for navigation tasks.

In summary, our work distinguishes itself by addressing Offline CRL in a goal-conditioned context without relying on data retention, and by introducing a new benchmark to fill this gap.

## 3 Preliminaries

We outline the core concepts behind our approach: Markov Decision Processes (MDP), Offline Goal-Conditioned RL, Continual Reinforcement Learning (CRL), neural network subspaces, and Low-Rank Adaptation (LoRA).

**Markov Decision Process (MDP).** We define an MDP as $\mathcal{M} = (\mathcal{S}, \mathcal{A}, \mathcal{P}_\mathcal{S}, \mathcal{P}_\mathcal{S}^{(0)}, \mathcal{R}, \gamma)$, where $\mathcal{S}$ and $\mathcal{A}$ are the state and action spaces ; $\mathcal{P}_\mathcal{S} : \mathcal{S} \times \mathcal{A} \to \Delta(\mathcal{S})$ is the transition function; $\mathcal{P}_\mathcal{S}^{(0)}$ is the initial state distribution ; $\mathcal{R} : \mathcal{S} \times \mathcal{A} \times \mathcal{S} \to \mathbb{R}$ is a deterministic reward function ; and $\gamma \in (0, 1]$ is the discount factor. The agent's behavior is given by a parameterized policy $\pi_\theta : \mathcal{S} \to \Delta(\mathcal{A})$ and the goal is to learn $\theta_\mathcal{M}^*$ that maximizes the goal-reaching success rate $\sigma_\mathcal{M}(\theta)$.

**Offline Goal-Conditioned RL.** We extend the MDP with a goal space $\mathcal{G}$ by introducing an initial state-goal distribution $\mathcal{P}_{\mathcal{S},\mathcal{G}}^{(0)}$, a mapping $\phi : \mathcal{S} \to \mathcal{G}$, and a distance metric $d : \mathcal{G} \times \mathcal{G} \to \mathbb{R}^+$. The policy becomes $\pi_\theta : \mathcal{S} \times \mathcal{G} \to \Delta(\mathcal{A})$ and $\mathcal{R}(s_t, a_t, s_{t+1}, g) = \mathbb{1}\big(d(\phi(s_{t+1}), g) \leq \epsilon\big)$, where sparse rewards are given only when the goal is reached within a threshold $\epsilon$. Training uses a dataset $\mathcal{D} = \{(s, a, r, s', g)\}$ to optimize the policy for each new goal-conditioned task.

**Continual Reinforcement Learning (CRL).** In CRL, the agent learns over a sequence of tasks $\mathcal{T} = (T_1, \ldots, T_N)$, where each $T_k$ is either an MDP $\mathcal{M}_k$ or a pair $(\mathcal{M}_k, \mathcal{D}_k)$. Let $\theta_k$ denote the parameters after learning task $T_k$. In navigation, environment changes are of two types : *Topological changes* (affecting high-level strategies) ; *Kinematic changes* (affecting low-level control). We evaluate CRL methods with standard metrics : **PER :** $\frac{1}{N} \sum_{k=1}^N \sigma_{\mathcal{M}_k}(\theta_N)$ ; **BWT :** $\frac{1}{N} \sum_{k=1}^N \big(\sigma_{\mathcal{M}_k}(\theta_N) - \sigma_{\mathcal{M}_k}(\theta_k)\big)$ ; **FWT :** $\frac{1}{N} \sum_{k=1}^N \big(\sigma_{\mathcal{M}_k}(\theta_k) - \sigma_{\mathcal{M}_k}(\tilde{\theta}_k)\big)$ ; and the relative memory usage **MEM**.

**Subspace of Neural Networks.** A subspace is the convex hull of a finite set of anchor points $\{\theta_1, \theta_2, \ldots, \theta_k\} \subset \Theta$. Any $\theta$ in the subspace can be written as $\theta = \sum_{i=1}^k \alpha_i \theta_i$, $\alpha \in \Delta^k$ $(\alpha_i \geq 0, \sum_{i=1}^k \alpha_i = 1)$. This lower-dimensional representation allows efficient adaptation. Our method employs two distinct subspaces (e.g., separating path planning and locomotion) and uses an offline evaluation procedure (see Section 4.3) to decide when to expand them.

**Low-Rank Adaptation (LoRA).** LoRA adapts a pretrained weight matrix $W \in \mathbb{R}^{n \times m}$ by adding a low-rank update: $W' = W + \Delta W$, $\Delta W = AB$, $A \in \mathbb{R}^{n \times r}$, $B \in \mathbb{R}^{r \times m}$, $r \ll \min(n, m)$. Within our framework, LoRA generates new anchors for the subspaces (for $i \geq 2$, $\theta_i = A_i B_i$). Our ablation study (Section 5.5.2) demonstrates how this approach further enhances performance.

## 4 Hierarchical Subspace of Policies

We now provide a detailed description of our framework : Hierarchical Subspaces of Policies (`HiSPO`). *Section 4.1* introduces Hierarchical Imitation Learning, the backbone algorithm of our proposed approach. Next, *Section 4.2* provides a high-level overview of the core learning steps involved in HiSPO. We then cover subspaces extension in *Section 4.3*, and subspace exploration in *Section 4.4*. See *Algorithm 1* for a detailed pseudo-code about our method for learning a subspace of policies in an offline setting.

## 4.1 Hierarchical Imitation Learning

Hierarchical Imitation Learning (Gupta et al., 2019) is the backbone of our approach for any given task, by learning policies using a dataset of pre-collected expert episodes $\mathcal{D} = \left\{ (s_t^i, a_t^i, r_t^i, s_{t+1}^i, g^i) \right\}$. The overall hierarchical policy is parameterized by $\theta = (\theta_h, \theta_l)$, where $\theta_h$ governs the high-level policy and $\theta_l$ controls the low-level one. This structure allows to break down complex tasks into simpler ones, through long-term planning and short-term actions .

- **High-Level Policy Training :** The high-level policy is trained to predict a sub-goal $\phi(s_{t+k})$, where $k$ is the *waystep* hyperparameter determining how far into the future the sub-goal is :

$$\mathcal{L}_{\mathcal{D}}^h(\theta_h) = \mathbb{E}_{(s_t^i, s_{t+k}^i, g^i) \sim \mathcal{D}} \left[ -\log(\pi_{\theta_h}^h(\phi(s_{t+k}^i)|s_t^i, g^i)) \right]$$

- **Low-Level Policy Training :** The low-level policy $\pi^l$ is trained to execute actions that take the agent towards the sub-goals proposed by the high-level policy :

$$\mathcal{L}_{\mathcal{D}}^l(\theta_l) = \mathbb{E}_{(s_t^i, a_t^i, s_{t+1}^i, \phi(s_{t+k}^i)) \sim \mathcal{D}} \left[ -\log(\pi_{\theta_l}^l(a_t|s_t^i, \phi(s_{t+k}^i))) \right]$$

- **Hindsight Experience Replay (HER) (Andrychowicz et al., 2017; Packer et al., 2021) :** We perform data augmentation using HER, which relabels the goal of a given transition with the goal representation of a future state within the same trajectories considered.

## 4.2 HiSPO Learning Algorithm : Overview

The `HiSPO` Learning Algorithm manages the hierarchical policies through distinct subspaces, each specializing to different aspects of task adaptation. This division promotes both efficiency and scalability, allowing our framework to handle diverse and sequential tasks in an offline setting.

**Initial Anchor Training :** We begin by training the initial anchor parameters $\theta_1^h \in \Theta^h$ and $\theta_1^l \in \Theta^l$ on the first task $T_1$. The anchor weights $\alpha_1^h \in \Delta^1$ and $\alpha_1^l \in \Delta^1$ are set to (1), indicating complete reliance on the initial anchors. This respectively establishes the foundational subspaces for high-level and low-level policies, later expanded for new tasks.

**Training on Subsequent Tasks :** For each new task $T_k$ and for each of the two considered subspaces, the algorithm performs the following steps to adapt the learning policy :

1. **Subspaces Extension :** We introduce parameters $\theta_{N^h+1}^h \in \Theta^h$ and $\theta_{N^l+1}^l \in \Theta^l$, and initialize the new anchor weights $\alpha_{\text{curr}}^h$ and $\alpha_{\text{curr}}^l$. These new anchors and anchors weights are then learned from the dataset $\mathcal{D}_k$ using Hierarchical Imitation Learning.

2. **Previous Subspaces Exploration and Evaluation :** We explore different anchor weights by sampling from a Dirichlet distribution with equal weights, to uniformly search over the previous subspace. Each of the sampled configuration is evaluated on a few batches from the new task's dataset $\mathcal{D}_k$, and the one minimizing the loss is selected as a representative of the previous subspace.

3. **Subspaces Adaptation Decision:** We compare the loss of the extended subspace ($L_{\text{curr}}$) with that of the previous subspace ($L_{\text{prev}}$) given a criterion $\epsilon > 0$. Considering positive losses, if $L_{\text{prev}} \leq (1 \pm \epsilon) \cdot L_{\text{curr}}$, we prune the new anchor, retaining the previous subspace configuration. Otherwise, we retain the lately learned anchor, effectively accommodating the subspace.

## 4.3 Extending a Single Subspace

Initially, $\theta_{N+1}$ is randomly initialized, and anchor scores $\hat{\alpha}_{\text{curr}} = (0, \ldots, 0)$ are set to zeros. During training, the softmax function is applied to the anchor scores, yielding the anchor weights $\alpha_{\text{curr}}$. This step ensures that the weights are positive and sum to one, providing differentiable control over how much each anchor contributes to the final policy. The learning process proceeds by updating parameters over the dataset, using mini-batches $\mathcal{B} \sim \mathcal{D}_{N+1}$, by considering the sampled weights contributions $\alpha_{\text{curr}}$ [1] :

$$(\hat{\alpha}_{\text{curr}}, \theta_{N+1}) \leftarrow (\hat{\alpha}_{\text{curr}}, \theta_{N+1}) - \eta \nabla \mathcal{L}_{\mathcal{B}} \left( \sum_{i=1}^{N+1} \alpha_{\text{curr},i} \cdot \theta_i \right)$$

---

[1]In contrast to CSP (Gaya et al., 2023) which relies on randomly sampling anchor weights upon learning anchors, we sample weights around $\alpha_{\text{curr}}$ with a fix *std.* of 0.1 to ensure both efficiency and parametric diversity.

In practice, whenever a new anchor is added, the anchor weights for previous tasks are extended by appending a zero to the weight vector. This ensures that the dimensionality of weight vectors is consistent across all tasks : $\alpha_i \leftarrow (\alpha_i, 0), \quad \forall i \in \{1, \ldots, N\}$ .

This approach allows the model to leverage knowledge from previously learned tasks while adjusting to the specific requirements of the new one. By adding the new anchor, the subspace is expanded, enabling the model to handle a broader range of tasks without forgetting previous skills.

## 4.4 Exploring a Single Subspace

After training the new anchor, we evaluate whether it should be kept or pruned. This decision is based on a comparison between the losses of the extended subspace (with the new anchor) and the previous subspace (without the new anchor). To compute the current subspace loss, we use $\alpha_{\text{curr}}$, and $L_{\text{curr}} = \mathcal{L}_{\mathcal{D}_k}\left(\sum_{i=1}^{N+1} \alpha_{\text{curr},i} \cdot \theta_i\right)$. For the previous one, this would involve finding $\alpha_{\text{prev}} = \arg\min_\alpha \mathcal{L}_{\mathcal{D}_k}\left(\sum_{i=1}^{N} \alpha_i \cdot \theta_i\right)$. However, in practice, performing a full optimization over $\alpha$ can be computationally expensive. Instead, we sample weights from a Dirichlet distribution over the simplex $\Delta^N$, providing a computationally efficient approximation to the optimization problem, and we compute the losses : $\alpha' \sim \text{Dir}(\Delta^N)$, $L'_{\text{prev}} = \mathcal{L}_{\mathcal{D}_k}\left(\sum_{i=1}^{N} \alpha'_i \cdot \theta_i\right)$, $L_{\text{prev}} = \min_{\alpha'} L'_{\text{prev}}$. Once both losses are computed, if the previous subspace loss $L_{\text{prev}}$ is within an acceptable range of the current subspace loss $L_{\text{curr}}$, the new anchor $\theta_{N+1}$ is pruned, and the anchor weights are reverted to the best previous configuration $\alpha_{\text{prev}}$. Specifically : $L_{\text{prev}} \leq (1 \pm \epsilon) \cdot L_{\text{curr}}$ . On the other hand, if the extended subspace performs significantly better, the subspace is retained, and the weights $\alpha_{\text{curr}}$ are kept.

---

**Algorithm 1** Offline Learning of a Subspace of Policies

---

**Require:** Stream $\mathcal{T}$ ; Sample size $S$ .
**Require:** E epochs ; Learning rate $\eta$ ; Criterion $\epsilon$ .
1: **A. Train initial anchors :**
2: Initialize anchor $\theta_1 \sim \Theta$ and weights $\alpha_1 \leftarrow (1)$
3: **for** $epoch = 1$ to E **do**
4:     With $\mathcal{B} \sim \mathcal{D}_1 : \theta_1 \leftarrow \theta_1 - \eta\nabla\mathcal{L}_{\mathcal{B}}(\alpha_{1,1} \cdot \theta_1)$
5: **B. Train subsequent anchors :**
6: **for** $k = 2$ to $\texttt{len}(\mathcal{T})$ **do**
7:     Consider the current anchors $\theta_1, \ldots, \theta_N \in \Theta$
8:     **B.1. Train $k$-th anchor :**
9:     Initialize anchor $\theta_{N+1} \sim \Theta$ and scores $\hat{\alpha}_{\text{curr}} \leftarrow (0)$
10:     **for** $epoch = 1$ to E **do**
11:         **for** mini-batch $\mathcal{B} \sim \mathcal{D}_k$ **do**
12:             - $\alpha_{\text{curr}} \sim \texttt{softmax}(\hat{\alpha}_{\text{curr}})$
13:             - $\theta_{N+1} \leftarrow \theta_{N+1} - \eta\nabla\mathcal{L}_{\mathcal{B}}(\sum_{i=1}^{N+1} \alpha_{\text{curr},i} \cdot \theta_i)$
14:             - $\hat{\alpha}_{\text{curr}} \leftarrow \hat{\alpha}_{\text{curr}} - \eta\nabla\mathcal{L}_{\mathcal{B}}(\sum_{i=1}^{N+1} \alpha_{\text{curr},i} \cdot \theta_i)$
15:     **B.2. Evaluate current subspace** (*Section 4.3*) :
16:     - $\alpha_{\text{curr}} \leftarrow \texttt{softmax}(\hat{\alpha}_{\text{curr}})$
17:     - $L_{\text{curr}} \leftarrow \mathcal{L}_{\mathcal{D}_k}\left(\sum_{i=1}^{N+1} \alpha_{\text{curr},i} \cdot \theta_i\right)$
18:     **B.3. Evaluate previous subspace** (*Section 4.4*) :
19:     - $\{\alpha'^{(s)}\}_{s=1}^{S} \sim \text{Dirichlet}(\mathbf{1}_N)$
20:     - $\alpha_{\text{prev}} \leftarrow \arg\min_{\alpha'^{(s)}} \mathcal{L}_{\mathcal{D}_k}\left(\sum_{i=1}^{N} \alpha'_i \cdot \theta_i\right)$
21:     - $L_{\text{prev}} \leftarrow \mathcal{L}_{\mathcal{D}_k}\left(\sum_{i=1}^{N} \alpha_{\text{prev},i} \cdot \theta_i\right)$
22:     **B.4. Criterion based adaptation decision :**
23:     **if** $L_{\text{prev}} \leq (1 \pm \epsilon) \cdot L_{\text{curr}}$ **then**
24:         **Pruning :** $\alpha_k \leftarrow \alpha_{\text{prev}}$, discard $\theta_{N+1}$
25:     **else**
26:         **Extending :** $\alpha_k \leftarrow \texttt{softmax}(\hat{\alpha}_{\text{curr}})$, keep $\theta_{N+1}$

---

## 5 EXPERIMENTS

Our experiments aim to address the following questions : How does `HiSPO` compare to relevant baselines in terms of performance metric and memory savings (Section 5.3) ? How well does it avoid forgetting and ensure generalization (Section 5.4) ? To go further, we explore through ablation studies (Section 5.5) : First, we explain the development and reasons behind `HiSPO`, demonstrating how it overcomes limitations of existing subspace methods ; Secondly, we present how *Low-Rank Adaptation* can improve memory savings of `HiSPO`, introducing the `HiLOW` framework ; Lastly, we formulate a *Probably Approximately Correct* selection criterion to avoid running subspace extensions, thus targeting *Zero-Shot Transfer Learning*.

### 5.1 ENVIRONMENTS & TASK STREAMS

We consider multiple scenarios designed to test the ability to adapt and transfer knowledge between navigation tasks. The experiments span two types of environments : classical maze benchmarks from Gymnasium (Lazcano et al., 2023) and video game environments implemented in Godot (Godot, 2020). Details about all environments and the tasks are provided in the *Appendix A*.

The classical maze environments , PointMaze and AntMaze, are well-known in deep learning but less explored in the CRL. We introduce a novel use of those by customizing datasets and environments from Minari (Younis et al., 2024) to create task variations such as shifting map or permuting actions. We also introduce 3D navigation environments in Godot, *SimpleTown* and *AmazeVille*, which feature raycast depthmaps, topological changes across task streams and human-authored datasets, reflecting the evolving nature of simulators such as in the video game industry. We assess the performance of the different approaches on a diverse set of task streams, with randomly generated sequences.

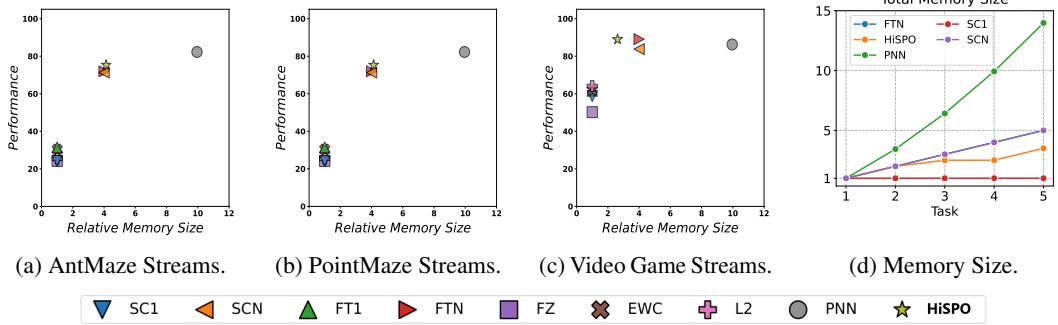

(a) AntMaze Streams.     (b) PointMaze Streams.     (c) Video Game Streams.     (d) Memory Size.

Figure 2: **Performance vs. Relative Memory Size.** The figure shows the average performance w.r.t. memory size of different CRL methods over streams from the defined environments. **HiSPO** (star) demonstrates **high performance** with **moderate memory usage**. Notably as show the Figure **(d)**, runs on random AntMaze tasks, our method is scalable and the resulting subspaces grow sublinearly.

### 5.2 CONTINUAL REINFORCEMENT LEARNING BASELINES

We compare our method to CRL strategies relevant to our setting, as described in *Section 3*. All baselines are built on a Hierarchical Imitation Learning backbone and detailed in *Appendix A.4*.

The **Naive Strategy (SC1)** trains a policy from scratch on the latest dataset and applies it to all tasks, while the **Expanding Naive Strategy (SCN)** saves a new policy for each task. The **Finetuning Strategy (FT1)** adapts a single policy, while the **Expanding Finetuning Strategy (FTN)**. The **Freeze Strategy (FRZ)** trains a policy on the first task and applies as it is to all the following tasks. More advanced methods include **L2-Regularization (L2)** (Kumar et al., 2023), which adds a penalty to the loss function according to previous weights, and **Elastic Weight Consolidation (EWC)** (Kirkpatrick et al., 2017), which improves L2 by penalizing important weights using the Fisher Information. **Progressive Neural Networks (PNN)** (Rusu et al., 2016) add new layers for each task, using lateral connections to leverage previous knowledge while avoiding interference.

## 5.3 Performance and Relative Memory Size

The trade-off between performance and memory usage is critical in CRL. *Figure 2* illustrates the average Performance (PER) according to the Relative Memory Size (MEM) of the baseline strategies and ours. `HiSPO` demonstrates high performance with moderate memory usage, outperforming or matching other methods in this balance.

In the **AntMaze streams**, our `HiSPO` method approaches the top-performing one **PNN** while using significantly less memory. The simple architectural strategies like **FTN** and **SCN** perform slightly below **HiSPO** with similar memory consumption. In contrast, weight regularization and naive methods (e.g., **EWC**, **FT1**, **FZ**) underperform. These results demonstrate that **HiSPO** effectively balances performance and resource use. In the **PointMaze streams**, **HiSPO** nearly matches the top-performing **PNN**, while also maintaining significantly lower memory usage. Simple architectural methods (**FTN** and **SCN**) show high task performance but require more memory compared to **HiSPO**. The weight regularization and naive strategies, as in AntMaze, fail to provide comparable performance, which highlights the advantage of **HiSPO** in memory-constrained tasks. In the **Video Game streams**, **HiSPO** surpasses **PNN** and **FTN** regarding memory usage while being as effective.

Overall, our method demonstrates good performance across diverse tasks while maintaining a lower memory usage, especially when compared to memory-heavy methods like **PNN**, which has an exponential memory cost according the number of tasks (see Figure 2). This balance makes **HiSPO** an efficient approach for continual reinforcement learning in resource-constrained environments.

## 5.4 Forgetting and Generalization

**Table 1** summarizes the Backward and Forward Transfer metrics for different methods across AntMaze, PointMaze, and Video Game streams. In general, architectural methods like **FTN**, **SCN**, **PNN** and  perform well in terms of **BWT**, as they can store parameters without overwriting previous ones, allowing them to avoid forgetting. On the other hand, weight regularization methods (**EWC**, **L2**) can struggle when task changes are more diverse, showing inconsistent BWT results. Regarding **FWT**, most methods exhibit either minimal or negative forward transfer, which highlights the challenge of knowledge transfer between tasks. While **FT1** and **FTN** demonstrate some positive forward transfer, **HiSPO** shows only modest improvements. This possibly indicates limitations in the adaptation strategy for further generalization. Although we acknowledge **HiSPO** does not excel in FWT, it maintains balanced performance across tasks, making it a robust option for effectively managing memory and performance in continual learning settings.

Table 1: **The Performance Metrics.** Backward Transfer (BWT) and Forward Transfer (FWT) across task streams.

| Method | AntMaze Streams | | |
|---|---|---|---|
| | PER ↑ | BWT ↑ | FWT ↑ |
| SC1 | 42.3 | -43.2 | 0.0 |
| SCN | 84.4 | **0.0** | 0.0 |
| FT1 | 54.4 | -40.4 | 3.6 |
| FTN | 86.8 | **0.0** | 3.6 |
| FZ | 35.9 | **0.0** | -50.3 |
| L2 | 48.6 | -34.4 | -2.5 |
| EWC | 49.7 | -39.0 | 3.3 |
| PNN | **89.3** | **0.0** | **3.8** |
| HiSPO | 87.7 | **0.0** | 2.0 |

## 5.5 Ablations

### 5.5.1 Online to Offline Subspace of Policies

**Adapting CSP to Offline CRL : `CSPO`.** The Continual Subspace of Policies (CSP) (Gaya et al., 2023) uses Soft Actor-Critic (SAC) (Haarnoja et al., 2018) with a Q-function for scoring. To adapt CSP for Offline CRL, we replace SAC with Hierarchical Imitation Learning (HBC) (see Table 2). However, Q-functions learned via Implicit Q-Learning (Kostrikov et al., 2021) may become nearly action-independent due to optimal expert data, limiting

Table 2: Ablations on AntMaze Streams.

| CSP Method | AntMaze Streams | |
|---|---|---|
| | PER ↑ | MEM ↓ |
| CSP | $42.4 \pm 5.7$ | $2.7 \pm 0.1$ |
| CSPO | $77.6 \pm 3.6$ | $2.3 \pm 0.5$ |
| HiSPO | $\mathbf{80.1 \pm 1.1}$ | $\mathbf{2.2 \pm 0.6}$ |

their offline effectiveness. CSPO overcomes this by employing a loss-based selection criterion that directly minimizes the loss relative to expert behavior while maintaining a single subspace.

**From One Subspace to Two : `HiSPO`.** `HiSPO` splits policy parameters into two subspaces—one for high-level and one for low-level policies—enabling fine-tuned updates and preventing unnecessary expansion of the high-level subspace. Our experiments show that `HiSPO` achieves performance comparable to CSPO while saving memory. Notably, CSPO adapts only to contexts similar to those already encountered.

### 5.5.2   LOW-RANK SUBSPACE OF POLICIES : `HiLOW`

Low-Rank Adaptation (LoRA) enables efficient parameter updates by approximating changes with low-rank matrices. Comparing our method with (`HiLOW`) and without (`HiSPO`) LoRA subspaces, we find that incorporating such adaptors allows for smaller, more efficient updates when adapting to new tasks.

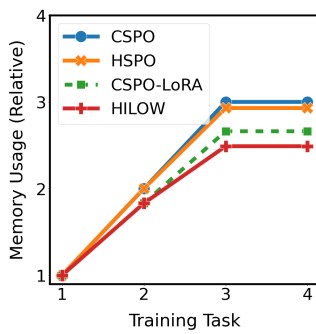

We believe future work could explore the interpretability of policy adaptations and refine the relationship between LoRA's rank and the nature of environmental changes, which could be interesting for industrial applications, and notably unsupervised hyperparameters tuning.

Figure 3: **Memory Usage on a AntMaze Stream**.

### 5.5.3   ZERO-SHOT TRANSFER LEARNING

Standard subspace extension expands the subspace and compares losses, which is computationally costly. To improve efficiency, we propose an empirical *Probably Approximately Correct* (PAC) criterion that evaluates the existing subspace without expansion.

Let $d : \mathcal{A} \times \mathcal{A} \to \mathbb{R}^+$ be a comparison function between data $(s, a) \in \mathcal{D}$ and policy outputs $\hat{a}$. Our criterion requires that, with probability at least $1 - \delta$, the current subspace's outputs are within an acceptable range $\epsilon$. Formally, we require $\mathbb{E}_{(s,a)\sim\mathcal{D},\, \hat{a}\sim\pi^*(\cdot|s)}\Big[\mathbb{1}\big(d(a, \hat{a}) \le \epsilon\big)\Big] \ge 1 - \delta$, where $\pi^*$ is the best policy from the previous subspace. The thresholds $\epsilon$ and $\delta$ control the acceptable deviation and confidence level.

For high-level navigation policies (with $\mathcal{G}$ a metric space), these parameters intuitively measure subspace capacity. This criterion enables zero-shot transfer learning without subspace expansion and paves the way for future work on automated hyperparameter tuning and quantifying subspace fitting.

## 6   DISCUSSION

In this work, we introduced `HiSPO`, a novel framework that combines hierarchical imitation learning with policy subspaces for offline continual reinforcement learning. Our results show that it effectively balances performance and memory usage across diverse tasks and environments, including classical mazes and 3D video games. By using separate subspaces for high-level and low-level policies, it efficiently adapts to new tasks while avoiding forgetting. Compared to other methods, `HiSPO` offers a strong trade-off between adaptability and resource efficiency.

Future work could study to which extent `HiSPO` can scale to more complex CRL settings, e.g. with chaotic task streams, or radiccaly different environments. Additionally, while it performs well with expert data, scenarios with imperfect expert trajectories could pose challenges. Towards such settings, improved subspace evaluations procedures could be studies, e.g. integrating *Inverse Reinforcement Learning* (IRL) (Arora & Doshi, 2021; Ho & Ermon, 2016), to provide better scoring of sampled policies. Exploring adaptive ranks during subspace extension could enhance the framework's flexibility and scalability with task complexity. Overall, `HiSPO` makes a significant step toward addressing offline continual learning challenges.

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

## A  TASK STREAMS DETAILS

### A.1  ENVIRONMENTS

#### A.1.1  MUJOCO MAZE ENVIRONMENTS

We consider two sets of environments from the Gymnasium framework Lazcano et al. (2023) : PointMaze and AntMaze. They are considered due to their complexity and the availability of datasets from D4RL Fu et al. (2020), which provide a standardized set of tasks to evaluate CRL algorithms.

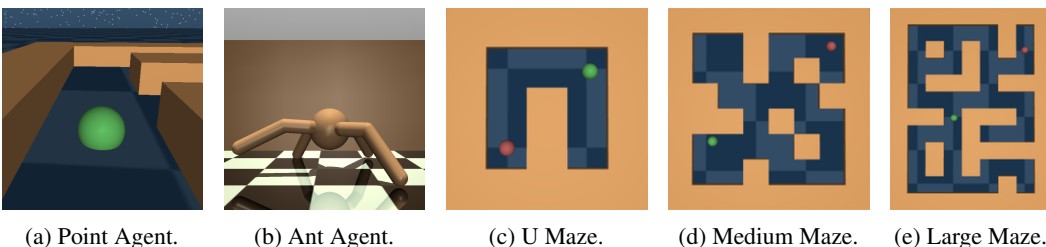

(a) Point Agent.    (b) Ant Agent.    (c) U Maze.    (d) Medium Maze.    (e) Large Maze.

Figure 4: All U (size $= 5 \times 5$), M (size $= 8 \times 8$), and L (size $= 12 \times 9$) mazes provide a sparse reward with a value of $1$ when the agent is within a $0.5$ unit radius to the goal. The **Point Agent** is a point mass controlled by applying forces in two dimensions, allowing the agent to move freely across the plane towards a goal location. In contrast the **Ant Agent** is a more complex articulated quadruped robot. It is controlled through the application of torques to its joints.

#### A.1.2  VIDEO GAME NAVIGATION ENVIRONMENTS

While PointMaze and AntMaze environments were simple to setup and allowed us to quickly generate datasets, as to our knowledge there are no CRL datasets for navigation, they are primarily focused on assessing the impact of changes in agent dynamics, such as action transformations. These environments are expressive but lack features needed to fully understand how topographic variations affect an agent. To bridge this gap, We introduce a video-game like 3D navigation environments, implemented on Godot (Godot (2020)), that offer diverse mazes with more explainable spatial challenges. They allow us to explore the influence of environmental structures on agent performance.

There are two families of mazes : **SimpleTown**, which mazes are relatively simple, with a size of $30 \times 30$ meters. The starting positions are randomly sampled on one side, and the goal positions are on the other side ; **AmazeVille**, which mazes are more challenging, with a size of $60 \times 60$ meters. They have a finite set of start and goal positions, and include two subsets of maps : some with high blocks, *i.e.* not jumpable obstacles ; others with low blocks, *i.e.* jumpable ones.

| Observation Feature | Size | Type | Observation Feature | Size | Type |
|---|---|---|---|---|---|
| Agent Position | 3 | float | Goal Position | 3 | float |
| Agent Orientation | 3 | float | Agent Velocity | 3 | float |
| RGB Image | $3 \times 64 \times 64$ | float | Depth Image | $11 \times 11$ | float |
| Floor Contact | 1 | bool | Wall Contact | 1 | bool |
| Goal Contact | 1 | bool | Timestep | 1 | int |
| Up Direction | 3 | float | - | - | - |

Table 3: **(Godot) Available observation features.** The maximum number of features an observation may have is $12440$, if it were to use all the available ones. The position information correspond to the $(x, y, z)$ coordinates in meters. The agent orientation is its angle in radian according to the vertical axis. The velocity is provided in meters per second. The RGB images corresponds to the visualization of the environment from the agent's point field of view. The depth image is obtained using $11 \times 11$ raycasts from the agent position to the visible nearest obstacles.

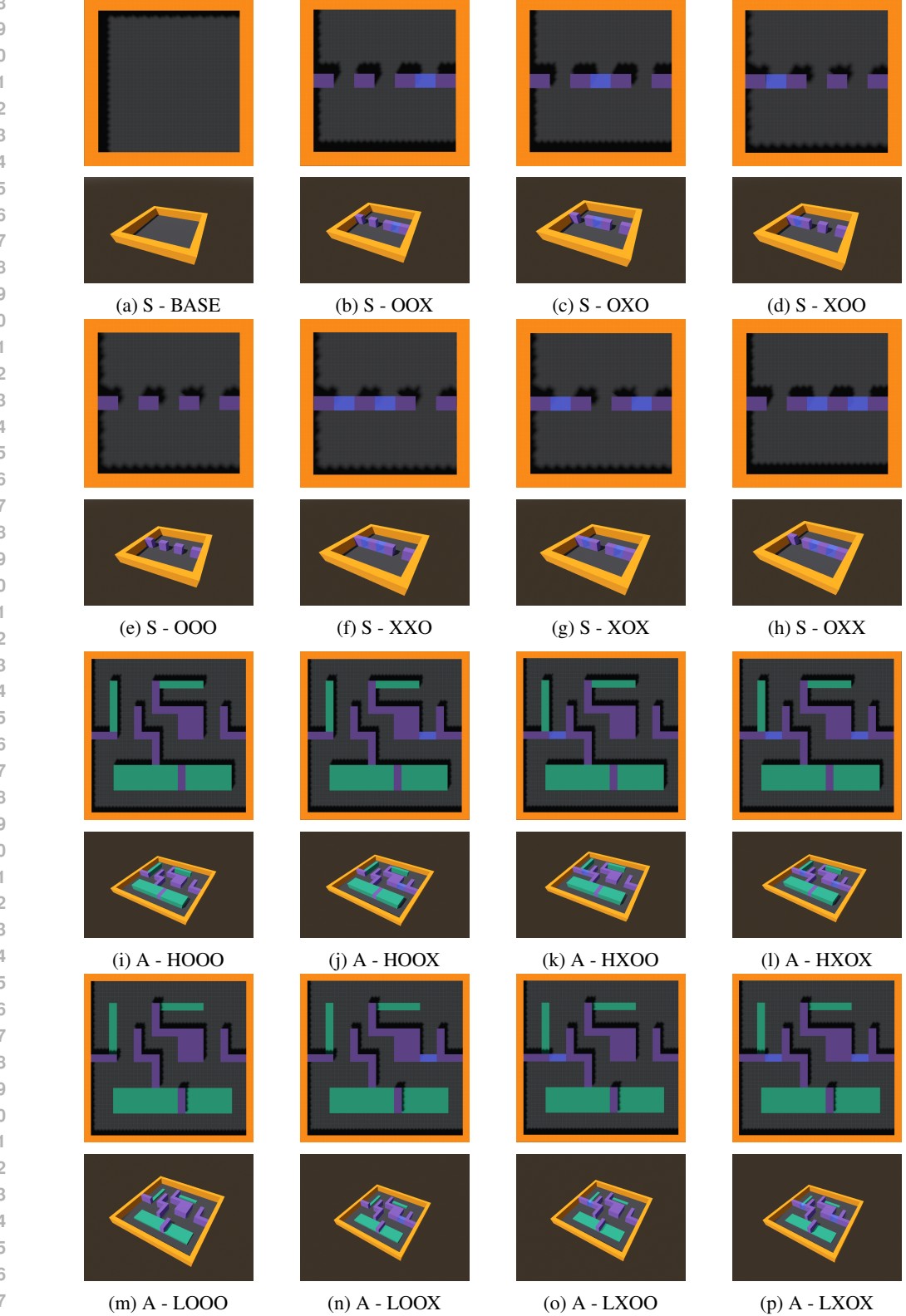

Figure 5: **The SimpleTown (S) and the AmazeVille (AH, AL) environments :** The naming indicate whether specific doors are open (O) or not (X), and if movable green blocks are in high positions (H) or low positions (L), providing a clear way to distinguish between different maze configurations.

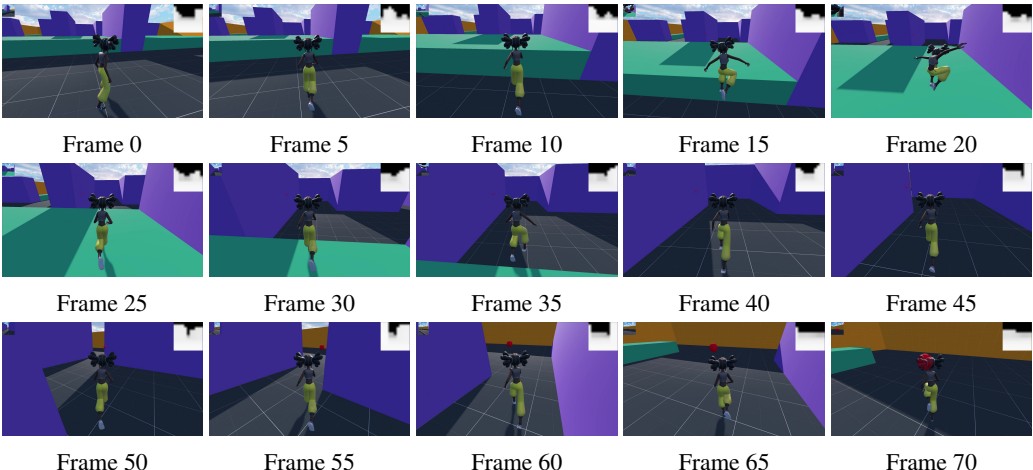

Figure 6: **Visualization of a Human-Generated Trajectory on A - LOOO.**

## A.2    TASKS

We design a variety of tasks within each environment to evaluate the agent's adaptability to different scenarios. For both PointMaze and AntMaze environments, we consider five task variations :

- Normal (N) : The standard task with no changes to actions or observations.
- Inverse Actions (IA) : Opposing values of the action features.
- Inverse Observations (IO) : Opposing values of the observation features.
- Permute Actions (PA) : clockwise permutation of the actions features.
- Permute Observations (PO) : Clockwise permutation of the observation features.

For the Godot-based environments (SimpleTown and AmazeVille), we simply use the mazes provided without additional modifications. The inherent complexity of these mazes, including variations in obstacle placement, already presents a significant challenge for the learning algorithms.

## A.3    DATASETS

For the PointMaze and AntMaze environments, we employed datasets from D4RL, each comprising 500 episodes per task across different maze configurations. Due to the straightforward nature of the task transformations, we effectively adapted the original datasets by applying these modifications and developed corresponding environment wrappers for seamless integration within the Gym framework.

The trajectories visualized in Figures 7 and 8 illustrate not only the richness and diversity of the collected data but also the complexity of the tasks that agents must navigate. These trajectories highlight a range of behaviors, from straightforward goal-reaching paths to more intricate maneuvers required to overcome environmental obstacles.

In the Godot-based environments, data was sampled manually over approximately 10 hours, resulting in 100 episodes for each AmazeVille maze and 250 episodes for each SimpleTown maze.

## A.4    TASK STREAMS

A task stream refers to a sequence of environments and corresponding datasets that an agent learns from over time. Each task in the stream introduces new environmental variations, changes in dynamics, or modifications to the observation and action spaces, simulating the possible evolving challenges in real-world scenarios. They may build upon previously learned skills, testing both short-term adaptability and long-term memory retention. We consider several classical metrics, namely : Performance (PER), Backward Transfer (BWT), Forward Transfer (FWT), and Relative Memory Size (MEM). These metrics enable us to assess the agent's continual learning capabilities by evaluating its ability to generalize across tasks, preserve learned knowledge, while being scalable.

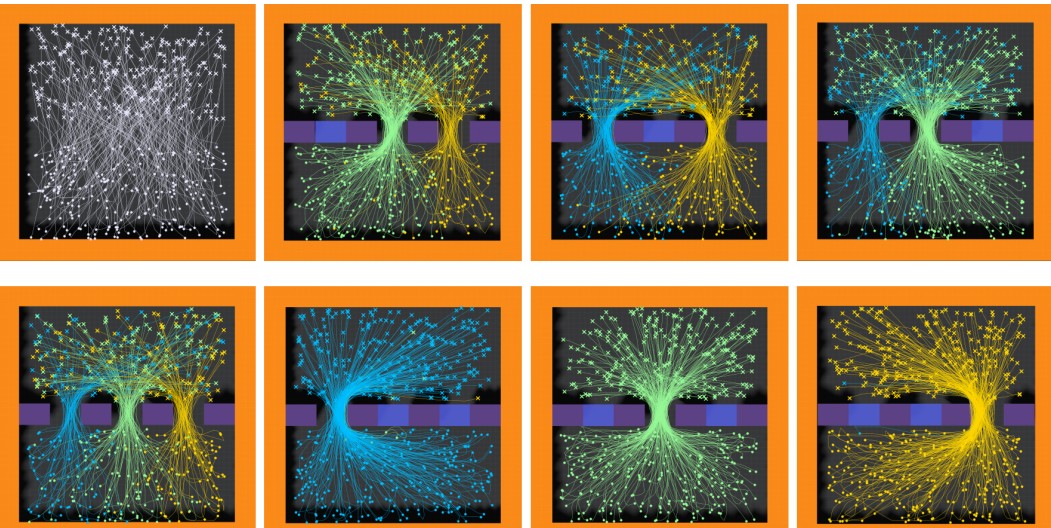

Figure 7: SimpleTown Trajectories (Staring Position are on the bottom).

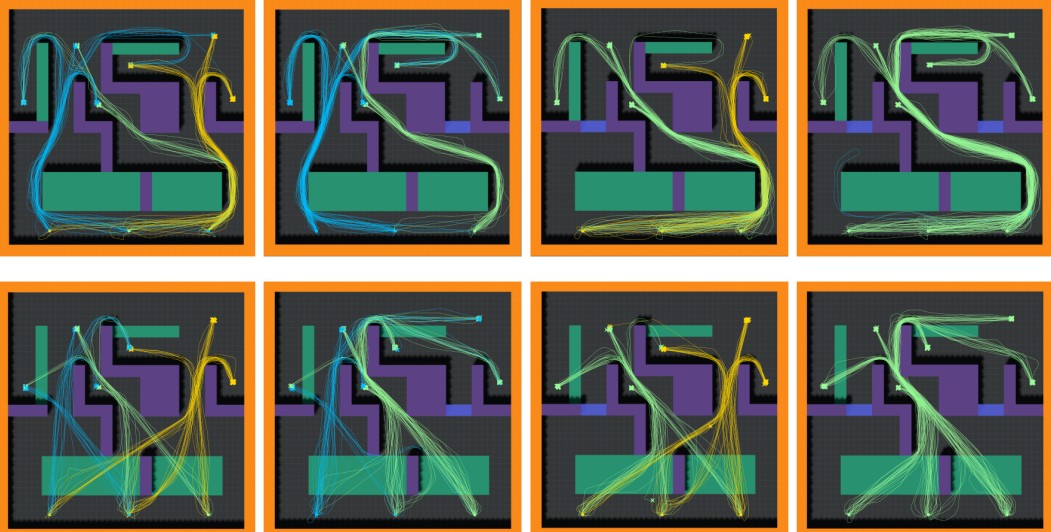

Figure 8: AmazeVille Trajectories (Staring Position are on the bottom).

Here are the AntMaze streams (`maze-task` $[n_{\text{episodes}}]$) :

- `1 : U-N[500]` $\to$ `L-N[500]` $\to$ `U-PO[500]` $\to$ `M-IO[500]`
- `2 : U-PA[500]` $\to$ `M-PO[500]` $\to$ `M-N[500]` $\to$ `M-N[500]`

Here are the PointMaze streams (`maze-task` $[n_{\text{episodes}}]$) :

- `1 : U-N[500]` $\to$ `L-N[500]` $\to$ `U-PO[500]` $\to$ `M-IO[500]`
- `2 : U-PA[500]` $\to$ `M-PO[500]` $\to$ `M-N[500]` $\to$ `M-N[500]`

Here are the Video Game streams (`maze` $[n_{\text{episodes}}]$) :

- `1 : HOOO[100]` $\to$ `HXOO[100]` $\to$ `LOOO[100]` $\to$ `LOOX[100]`
- `2 : LOOX[100]` $\to$ `HXOO[100]` $\to$ `HOOO[100]` $\to$ `LXOX[100]`

## B  BASELINES DETAILS

### B.1  GOAL-CONDITIONED OFFLINE REINFORCEMENT LEARNING ALGORITHMS

In both Imitation Learning and Hierarchical Imitation Learning algorithms, we will consider a MDP $\mathcal{M} = \left( \, \mathcal{S}, \, \mathcal{A}, \, \mathcal{P}_{\mathcal{S}}, \, \mathcal{P}_{\mathcal{S},\mathcal{G}}^{(0)}, \, \mathcal{R}, \, \gamma, \, \mathcal{G}, \, \phi, \, d \, \right)$, and a dataset of pre-collected trajectories $\mathcal{D} = \left\{ \, (s_t^i, a_t^i, r_t^i, s_{t+1}^i, g^i) \, \right\}$ sampled by one or many expert agents.

**Imitation Learning (BC) Ding et al. (2019).**   The BC algorithm is a simple framework to leverage a dataset of transitions $\mathcal{D}$ by running a supervised regression using a negative log-likelihood loss :

$$\mathcal{L}_{\mathcal{D}}(\theta) = \mathbb{E}_{(s_t^i, a, s_t^i, r, s_t^i, s_{t+1}^i, g^i) \sim \mathcal{D}} \left[ \, - log(\pi_\theta(a_t^i | s_t^i, g^i)) \, \right] \, , \text{ and } \, \theta_{\mathcal{D}}^* = \arg\min_{\theta \, \in \, \Theta} \mathcal{L}_{\mathcal{D}}(\theta) \qquad (1)$$

Moreover this algorithm benefit from using a HER (Figure 9) relabelling strategy. Indeed, as the trajectories have been sampled by an expert, if we consider a transition $(s_t^i, a_t^i, r_t^i, s_{t+1}^i, g^i) \in \mathcal{D}$ then we can also consider $(s_t^i, a_t^i, r_t^i, s_{t+1}^i, \phi(s_{t+k}^i))$ as also an expert generated transition. Thus, HER can be considered as a data augmentation technique, which is particularly effective in low data regime.

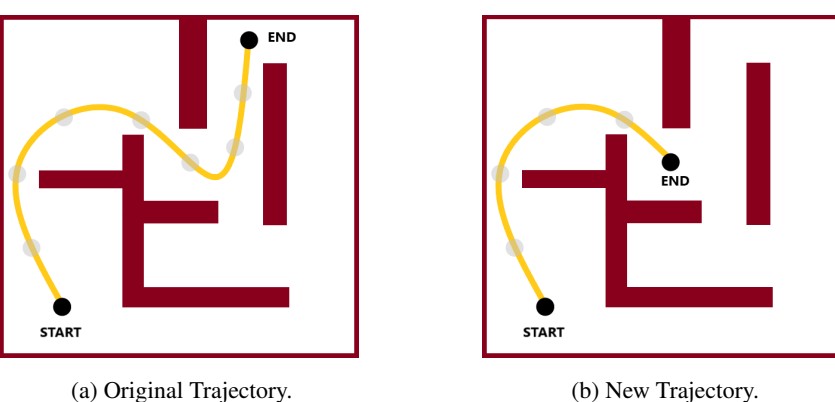

(a) Original Trajectory.                    (b) New Trajectory.

Figure 9: Hindsight Experience Replay (HER) Illustration.

**Hierarchical Imitation Learning (HBC) (Le et al., 2018; Gupta et al., 2019; Park et al., 2023).**
HBC leverages hierarchical structures so as to effectively handle the challenges associated with learning from offline datasets. This algorithm decomposes the navigation task into manageable sub-tasks using a high-level and a low-level policy.

Now, an end-to-end policy $\pi_\theta : \mathcal{S} \times \mathcal{G} \to \Delta(\mathcal{A})$ is divided into two distinct learnable components. First, a high policy $\pi_{\theta_h}^h : \mathcal{S} \times \mathcal{G} \to \Delta(\mathcal{G})$ aiming at selecting intermediate sub-goals that are strategically feasible stepping stones towards a final goal, thus simplifying the path finding task. Then, a low policy $\pi_{\theta_l}^l : \mathcal{S} \times \mathcal{G} \to \Delta(\mathcal{A})$ focused on generating the actions necessary to progress from the current state towards the sub-goal selected by the high policy. The optimization follows :

$$\mathcal{L}_{\mathcal{D}}^h(\theta_h) = \mathbb{E}_{(s_t^i, s_{t+k}^i, g^i) \sim \mathcal{D}} \left[ \, - log(\pi_{\theta_h}^h(\phi(s_{t+k}^i) | s_t^i, g^i))) \, \right] \, , \text{ and } \, \theta_h{}_{\mathcal{D}}^* = \arg\min_{\theta_h \, \in \, \Theta} \mathcal{L}_{\mathcal{D}}^h(\theta_h) \qquad (2)$$

$$\mathcal{L}_{\mathcal{D}}^l(\theta_l) = \mathbb{E}_{(s_t^i, a_t^i, s_{t+1}^i, s_{t+k}^i, g^i) \sim \mathcal{D}} \left[ \, - log(\pi_{\theta_l}^l(a_t | s_t^i, \phi(s_{t+k}^i))) \, \right] \, , \text{ and } \, \theta_l{}_{\mathcal{D}}^* = \arg\min_{\theta_l \, \in \, \Theta} \mathcal{L}_{\mathcal{D}}^l(\theta_l) \qquad (3)$$

Hence, given *way step* hyperparameter $k$, which determines the desired temporal distance of the sub-goals, the optimization for the high and low policies uses a common loss structure, adapted to suit their specific roles.

## B.2 CONTINUAL REINFORCEMENT LEARNING BASELINES

This section explores CRL baselines, designed to learn from a task stream $\mathcal{T}$, where each task $T_k$ consists of a MDP $\mathcal{M}_k = \left( \mathcal{S}_k, \mathcal{A}_k, \mathcal{P}_{\mathcal{S}_k}, \mathcal{P}_{\mathcal{S},\mathcal{G}_k}^{(0)}, \mathcal{R}_k, \gamma_k, \mathcal{G}, \phi_k, d_k \right)$ and a dataset of trajectories $\mathcal{D}_k = \left\{ (s_t^{k,i}, a_t^{k,i}, r_t^{k,i}, s_{t+1}^{k,i}, g^{k,i}) \right\}$. Interestingly, these strategies could be extended to a broader range algorithms, beyond goal-conditioned ones.

**Naive Learning Strategy or** *From Scratch* **(`SC1` & `SCN`).** In `SC1`, a single policy is learned from the latest dataset and then applied unchanged to all tasks. In `SCN`, a new policy is trained for each task, improving performance at the cost of a memory load.

---

**Algorithm 2** Naive Strategy

---

**Require:** learning rate $\eta$, number of epochs `E`, boolean `StorePolicies`
 1: **for** $k = 1$ to $N$ **do**
 2:      Initialize policy parameters $\theta_k$
 3:      **for** $epoch = 1$ to `E` **do**
 4:          **for** mini-batch $\mathcal{B}$ in $\mathcal{D}_k$ **do**
 5:              Update $\theta_k$ using gradient descent: $\theta_k \leftarrow \theta_k - \eta \nabla \mathcal{L}_{\mathcal{B}}(\theta_k)$
 6:      **if** `StorePolicies` **then** Store $\theta_k$
 7:      **else** $\theta_1 \leftarrow \theta_k$

---

**Freeze Strategy (`FZ`).** In the Freeze Strategy, a single policy is trained only on the first task and then applied without modification to all subsequent tasks.

---

**Algorithm 3** Freeze Strategy

---

**Require:** learning rate $\eta$, number of epochs `E`
 1: Initialize policy parameters $\theta_1$
 2: **for** $epoch = 1$ to `E` **do**
 3:      **for** mini-batch $\mathcal{B}$ in $\mathcal{D}_1$ **do**
 4:          Update $\theta_1$ using gradient descent: $\theta_1 \leftarrow \theta_1 - \eta \nabla \mathcal{L}_{\mathcal{B}}(\theta_1)$

---

**Finetuning Strategy (`FT1` & `FTN`).** The Finetuning Strategy involves adapting a policy learned from the initial task to each subsequent task, either by continuously updating a single policy (`FT1`) or by copying and then updating the policy for each new task (`FTN`), allowing for better task adaptation.

---

**Algorithm 4** Finetuning Strategy

---

**Require:** learning rate $\eta$, number of epochs `E`, boolean `StorePolicies`
 1: Initialize policy parameters $\theta_1$
 2: **for** $epoch = 1$ to `E` **do**
 3:      **for** mini-batch $\mathcal{B}$ in $\mathcal{D}_1$ **do**
 4:          Update $\theta_1$ using gradient descent: $\theta_1 \leftarrow \theta_1 - \eta \nabla \mathcal{L}_{\mathcal{B}}(\theta_1)$
 5: **for** $k = 2$ to $N$ **do**
 6:      **if** `StorePolicies` **then** $\theta_k \leftarrow \theta_{k-1}$
 7:      **else** $\theta_k \leftarrow \theta_1$
 8:      **for** $epoch = 1$ to `E` **do**
 9:          **for** mini-batch $\mathcal{B}$ in $\mathcal{D}_k$ **do**
10:              Update $\theta_k$ using gradient descent: $\theta_k \leftarrow \theta_k - \eta \nabla \mathcal{L}_{\mathcal{B}}(\theta_k)$
11:      **if** `StorePolicies` **then** Store $\theta_k$
12:      **else** $\theta_1 \leftarrow \theta_k$

---

**Elastic Weight Consolidation (`EWC`) Kirkpatrick et al. (2017).** This strategy has been designed to mitigate catastrophic forgetting in continual learning. It achieves this by selectively slowing down learning on certain weights based on their importance to previously learned tasks. This importance is measured by the Fisher Information Matrix, which quantifies the sensitivity of the output function to changes in the parameters.

`EWC` introduces a quadratic penalty to the loss function, constraining the parameters close to their values from previous tasks, where the strength of the penalty is proportional to each parameter's importance. This allows the model to retain performance on previous tasks while continuing to learn new tasks effectively.

However this method struggles for navigation tasks due to the penalty for updating parameters, making it difficult to adapt to tasks like inverse actions. This rigidity is problematic in complex environments where different tasks demand flexibility. As a result, `EWC` is limited in effectively handling tasks requiring greater adaptation.

---

**Algorithm 5** Elastic Weight Consolidation Strategy

---

**Require:** learning rate $\eta$, number of epochs E, elastic weight $\lambda$, Fisher Information Matrix $\mathcal{F}_0$
1: Initialize policy parameters $\theta$
2: **for** $k = 1$ to $N$ **do**
3:     **for** $epoch = 1$ to E **do**
4:         **for** mini-batch $\mathcal{B}$ in $\mathcal{D}_k$ **do**
5:             Compute standard loss : $\mathcal{L}_{\mathcal{B}}^{S}(\theta)$
6:             Compute EWC loss : $\mathcal{L}_{\mathcal{B}}^{EWC}(\theta) = \frac{\lambda}{2} \sum_{i=1}^{k-1} \mathcal{F}_i \cdot (\theta_i - \theta_{i,\text{old}})^2$
7:             Total loss : $\mathcal{L}_{\mathcal{B}}(\theta) = \mathcal{L}_{\mathcal{B}}^{S}(\theta) + \mathcal{L}_{\mathcal{B}}^{EWC}(\theta)$
8:             Update $\theta$ using gradient descent: $\theta \leftarrow \theta - \eta \nabla \mathcal{L}_{\mathcal{B}}(\theta)$
9:     Update Fisher Information Matrix $\mathcal{F}_k$
10:    Store current parameters to learn next ones $\theta_{k,\text{old}} \leftarrow \theta$

---

**L2-Regularization Finetuning (L2) Kumar et al. (2023).** This strategy also mitigates catastrophic forgetting by adding an L2 penalty to the loss, discouraging large weight changes during training. This helps preserve knowledge from previous tasks by promoting stability in the learned representations.

As with EWC, L2-regularization struggles in CRL for navigation tasks, especially when actions or dynamics change drastically. The method limits the network's flexibility by forcing small weight updates, making it difficult to adapt to tasks that require distinct actions for similar states, which is critical in evolving environments.

---

**Algorithm 6** L2-Regularization Finetuning Strategy

---

**Require:** learning rate $\eta$, number of epochs E, regularization strength $\lambda$
1: Initialize policy parameters $\theta$ with $\theta$
2: **for** $k = 1$ to $N$ **do**
3:     **for** $epoch = 1$ to E **do**
4:         **for** mini-batch $\mathcal{B}$ in $\mathcal{D}_k$ **do**
5:             Compute task-specific loss : $\mathcal{L}_{\mathcal{B}}^{S}(\theta)$
6:             Compute L2 regularization loss : $\mathcal{L}_{\mathcal{B}}^{L2}(\theta) = \lambda \|\theta - \theta_{\text{old}}\|^2$
7:             Total loss : $\mathcal{L}_{\mathcal{D}_k}(\theta) = \mathcal{L}_{\mathcal{B}}^{S}(\theta) + \mathcal{L}_{\mathcal{B}}^{L2}(\theta)$
8:             Update $\theta$ using gradient descent: $\theta \leftarrow \theta - \eta \nabla \mathcal{L}_{\mathcal{B}}(\theta)$

---

**Progressive Neural Networks (PNN) Rusu et al. (2016).** This framework introduce a new column layers for each task, freezing previous weights to preserve knowledge. Lateral connections allow feature transfer, leveraging prior experience while avoiding interference. PNNs effectively prevent catastrophic forgetting, but the model grows with each task, limiting scalability for many tasks or limited memory contexts.

---

**Algorithm 7** Progressive Neural Networks Strategy

---

**Require:** number of tasks $N$, learning rate $\eta$
1: Initialize first task column $C_1$ with random weights
2: Train $C_1$ on the dataset $\mathcal{D}_1$ for the first task
3: **for** $k = 2$ to $N$ **do** ▷ For each new task
4:     Create a new task-specific column $C_k$ with random weights
5:     Freeze weights in previous columns $C_1, C_2, \ldots, C_{k-1}$
6:     Add lateral connections from $C_1, \ldots, C_{k-1}$ to $C_k$
7:     Load task-specific dataset $\mathcal{D}_k$
8:     **for** each mini-batch $\mathcal{B}$ in $\mathcal{D}_k$ **do**
9:         Compute the outputs of previous columns $C_1, \ldots, C_{k-1}$
10:         Pass outputs through lateral connections to $C_k$
11:         Update the weights in $C_k$ using gradient descent
12:     Freeze the weights in column $C_k$ after training

---

**Continual Subspace of Policies (CSP) Gaya et al. (2023).** This strategy handles continual learning by maintaining a subspace of policy parameters that adapt as new tasks are learned. For each new task, a new anchor is added, allowing the model to combine parameters from previous tasks. CSP decides whether to extend or prune the subspace based on a critic, $W_\phi$, that evaluates the performance of anchor combinations.

---

**Algorithm 8** Continual Subspace of Policies (CSP)

---

1: **Input:** $\theta_1, \ldots, \theta_j$ (previous anchors), $\epsilon$ (threshold)
2: **Initialize:** $W_\phi$ (subspace critic), $\mathcal{B}$ (replay buffer)
3: **Initialize:** $\theta_{j+1} \leftarrow \frac{1}{j} \sum_{i=1}^{j} \theta_i$ (new anchor)
4: **for** $i = 1, \ldots, \mathcal{B}$ **do** ▷ // Grow the Subspace
5:     Sample $\alpha \sim \text{Dir}(\mathcal{U}(j+1))$
6:     Set policy parameters $\theta_\alpha \leftarrow \sum_{i=1}^{j+1} \alpha_i \theta_i$
7:     **for** $l = 1, \ldots, K$ **do**
8:         Collect and store $(s, a, r, s', \alpha)$ in $\mathcal{B}$ by sampling $a \sim \pi_{\theta_\alpha}(s)$
9:     **if** *time to update* **then**
10:         Update $\pi_{\theta_{j+1}}$ and $W_\phi$ using the SAC algorithm and the replay buffer $\mathcal{B}$
11:
12: Use $\mathcal{B}$ and $W_\phi$ to estimate: ▷ // Extend or Prune the Subspace
13:

$$\alpha^{\text{old}} \leftarrow \underset{\alpha \in \mathbb{R}_+^n, \|\alpha\|_1 = 1}{\arg\max} W_\phi(\alpha)$$

$$\alpha^{\text{new}} \leftarrow \underset{\alpha \in \mathbb{R}_+^{n+1}, \|\alpha\|_1 = 1}{\arg\max} W_\phi(\alpha)$$

14: **if** $W_\phi(\cdot, \alpha^{\text{new}}) > (1 + \epsilon) \cdot W_\phi(\cdot, \alpha^{\text{old}})$ **then**
15:     **Return:** $\theta_1, \ldots, \theta_j, \theta_{j+1}, \alpha^{\text{new}}$ ▷ // Extend
16: **else**
17:     **Return:** $\theta_1, \ldots, \theta_j, \alpha^{\text{old}}$ ▷ // Prune

---

## C    IMPLEMENTATION DETAILS

### C.1    ARCHITECTURES & HYPERPARAMETERS

We primarily followed prior work (Ghosh et al., 2023) for network architectures and hyperparameters. All environments used MLPs with layer normalization on hidden layers. Low-level policies had 256 hidden units, and high-level policies used 64. For HiSPO in AntMaze and Godot, we increased these to 300 and 70 respectively as, experimentally, low-rank adaptors performed better with larger initial models on more complex tasks. Dropout of 0.1 was applied to all hidden layers.

Input sizes were 31 for **AntMaze** (including position, goal, and features), 8 for **PointMaze**, and 133 for **Godot**. Output sizes were 8 for AntMaze and Godot, and 2 for PointMaze. Outputs were continuous for AntMaze and PointMaze, while Godot used both continuous and discrete outputs to simulate gamepad controls.

| Hyperparameter | AntMaze | PointMaze | AmazeVille | SimpleTown |
|---|---|---|---|---|
| Batch Size | 1024 | 1024 | 64 | 64 |
| Learning Rate | 3e-4 | | | |
| Way Steps (Sub-goal distance) | Umaze : 10
Medium : 15
Large : 15 | Umaze : 50
Medium : 25
Large : 25 | 10 | 3 |
| HER Sampling Temperature | 50.0 | Umaze : 100.0
Medium : 75.0
Large : 100.0 | 100.0 | 15.0 |

Table 4: Hyperparameter settings for AntMaze, PointMaze, and Godot environments.

### C.2    TRAINING DETAILS

For both the EWC and the L2 strategies, we experimented with five different regularization weights $\lambda \in \{ 1e\text{-}2, 1e\text{-}1, 1, 1e1, 1e2 \}$ and selected the *best* model in terms of performance for each task stream. Similarly, for HiSPO, we tested different acceptance values $\epsilon \in \{ 1e\text{-}2, 5e\text{-}2, 1e\text{-}1, 2.5e\text{-}1 \}$ to decide whether to prune or extend a subspace.

When using Hierarchical Imitation Learning, we also employed Hindsight Experience Replay (HER) for all environments, using an exponential sampling strategy guided by a temperature parameter to improve sample efficiency.

### C.3    COMPUTE RESOURCES

Training was conducted on a shared compute cluster using CPUs for all experiments, as the models are relatively small and the backbone algorithms do not require highly intensive operations typically associated with GPU use. This choice also allowed us to run more experiments in parallel, optimizing resource utilization. The compute cluster featured Intel(R) Xeon(R) CPU E5-1650 and Intel Cascade Lake 6248 processors. For most models, 4 cores per training were sufficient, but due to PNN's growing memory requirements, we allocated 6 cores for its experiments. Total training times across the defined streams of tasks ranged from 10 to 18 hours, depending on the complexity of the task stream and the run time of the considered CRL strategy.

# D  ADDITIONAL & DETAILED RESULTS

## D.1  HIERARCHICAL VS. NON-HIERARCHICAL POLICIES IN GOAL-CONDITIONED RL

*Table 5* compares Imitation Learning and Hierarchical Imitation Learning across the various maze environments. HBC consistently outperforms BC in both success rate and episode length, especially in complex environments like AmazeVille, where hierarchical decision-making is crucial for navigating diverse tasks and obstacles. In simpler environments like SimpleTown, the performance difference is minimal, as these tasks are easier to solve.

| Environment | Maze | Success Rate ↑ | | Episode Length ↓ | |
|---|---|---|---|---|---|
| | | BC | HBC | BC | HBC |
| **PointMaze** | Umaze | $99.2 \pm 1.4$ | $\mathbf{100.0} \pm \mathbf{0.0}$ | $68.4 \pm 10.9$ | $\mathbf{63.8} \pm \mathbf{6.2}$ |
| | Medium | $94.1 \pm 8.4$ | $\mathbf{99.5} \pm \mathbf{1.1}$ | $199.5 \pm 32.2$ | $\mathbf{172.0} \pm \mathbf{33.1}$ |
| | Large | $67.9 \pm 9.7$ | $\mathbf{95.0} \pm \mathbf{6.9}$ | $328.5 \pm 33.3$ | $\mathbf{282.5} \pm \mathbf{61.4}$ |
| **AntMaze** | Umaze | $76.7 \pm 8.5$ | $\mathbf{93.5} \pm \mathbf{5.4}$ | $422.0 \pm 75.9$ | $\mathbf{286.6} \pm \mathbf{48.8}$ |
| | Medium | $43.3 \pm 10.5$ | $\mathbf{68.8} \pm \mathbf{5.0}$ | $688.0 \pm 101.1$ | $\mathbf{519.1} \pm \mathbf{61.4}$ |
| | Large | $18.8 \pm 11.4$ | $\mathbf{32.8} \pm \mathbf{9.9}$ | $861.4 \pm 88.9$ | $\mathbf{816.8} \pm \mathbf{63.8}$ |
| **SimpleTown** | BASE | $94.8 \pm 5.0$ | $\mathbf{98.6} \pm \mathbf{2.0}$ | $52.7 \pm 3.6$ | $\mathbf{51.5} \pm \mathbf{2.6}$ |
| | OOO | $95.9 \pm 1.9$ | $\mathbf{97.3} \pm \mathbf{1.9}$ | $\mathbf{55.8} \pm \mathbf{2.2}$ | $56.0 \pm 2.5$ |
| | OOX | $92.6 \pm 4.8$ | $\mathbf{94.3} \pm \mathbf{3.2}$ | $60.6 \pm 2.3$ | $\mathbf{59.7} \pm \mathbf{4.0}$ |
| | OXO | $89.5 \pm 4.4$ | $\mathbf{91.6} \pm \mathbf{4.2}$ | $\mathbf{61.7} \pm \mathbf{1.9}$ | $62.8 \pm 1.2$ |
| | XOO | $\mathbf{94.0} \pm \mathbf{4.0}$ | $93.8 \pm 3.7$ | $\mathbf{59.3} \pm \mathbf{3.0}$ | $60.0 \pm 2.4$ |
| | XXO | $\mathbf{89.8} \pm \mathbf{7.2}$ | $84.2 \pm 5.3$ | $\mathbf{70.2} \pm \mathbf{2.5}$ | $72.6 \pm 1.7$ |
| | XOX | $90.1 \pm 5.7$ | $\mathbf{97.0} \pm \mathbf{2.3}$ | $61.4 \pm 2.5$ | $\mathbf{60.2} \pm \mathbf{1.8}$ |
| | OXX | $\mathbf{93.4} \pm \mathbf{4.3}$ | $91.3 \pm 3.0$ | $\mathbf{67.5} \pm \mathbf{0.9}$ | $69.5 \pm 1.6$ |
| **AmazeVille** | HOOO | $70.5 \pm 9.7$ | $\mathbf{88.8} \pm \mathbf{6.3}$ | $211.0 \pm 12.8$ | $\mathbf{182.5} \pm \mathbf{9.3}$ |
| | HOOX | $51.2 \pm 13.0$ | $\mathbf{78.6} \pm \mathbf{8.7}$ | $249.8 \pm 18.9$ | $\mathbf{226.0} \pm \mathbf{14.2}$ |
| | HXOO | $60.4 \pm 15.8$ | $\mathbf{94.8} \pm \mathbf{4.7}$ | $228.3 \pm 19.8$ | $\mathbf{190.8} \pm \mathbf{9.1}$ |
| | HXOX | $46.5 \pm 9.9$ | $\mathbf{75.9} \pm \mathbf{5.2}$ | $273.7 \pm 11.9$ | $\mathbf{240.8} \pm \mathbf{4.7}$ |
| | LOOO | $49.6 \pm 3.5$ | $\mathbf{75.0} \pm \mathbf{7.1}$ | $221.9 \pm 6.0$ | $\mathbf{172.2} \pm \mathbf{18.0}$ |
| | LOOX | $59.9 \pm 7.2$ | $\mathbf{82.9} \pm \mathbf{6.3}$ | $225.9 \pm 12.2$ | $\mathbf{174.8} \pm \mathbf{9.6}$ |
| | LXOO | $47.0 \pm 5.8$ | $\mathbf{75.9} \pm \mathbf{6.3}$ | $222.8 \pm 8.3$ | $\mathbf{169.3} \pm \mathbf{13.6}$ |
| | LXOX | $60.1 \pm 8.8$ | $\mathbf{95.6} \pm \mathbf{4.6}$ | $221.3 \pm 14.8$ | $\mathbf{159.9} \pm \mathbf{10.1}$ |

Table 5: **Performance of BC and HBC across baseline environments (average over 8 seeds).** HBC consistently outperforms BC in both success rate and episode length metrics across most environments. In some of the SimpleTown environments, the differences between HBC and BC are negligible, as these tasks are easier to learn and provide limited room for improvement.

Given its efficiency in managing complex environments, HBC was chosen as the backbone for the HiSPO framework. By separating high-level and low-level subspaces, HiSPO further enhances task adaptation while avoiding unnecessary model expansion, making it well-suited for continual learning in dynamic, complex settings.

## D.2 Hierarchical vs. Non-Hierarchical Policies in Goal-Conditioned CRL

*Table 6* consistently demonstrate that HBC improves over BC, notably in terms of performance (PER) across all CRL baselines tested on both the PointMaze-1 and AntMaze-1 task streams. The most notable improvements are observed in sophisticated methods like FTN, SCN, and PNN, where HBC achieves near-perfect scores, such as 99.4 in PointMaze-1's PNN compared to BC's 96.9.

| Task Stream | CRL Method | PER ↑ | | MEM ↓ | |
|---|---|---|---|---|---|
| | | BC | HBC | BC | HBC |
| PointMaze-1 | EWC | $53.7 \pm 13.7$ | $\mathbf{55.1} \pm \mathbf{2.9}$ | $\mathbf{1.0} \pm \mathbf{0.0}$ | $1.1 \pm 0.0$ |
| | FT1 | $\mathbf{61.4} \pm \mathbf{16.4}$ | $50.0 \pm 2.8$ | $\mathbf{1.0} \pm \mathbf{0.0}$ | $1.1 \pm 0.0$ |
| | FTN | $95.0 \pm 0.9$ | $\mathbf{99.1} \pm \mathbf{0.8}$ | $\mathbf{4.0} \pm \mathbf{0.0}$ | $4.3 \pm 0.0$ |
| | FZ | $\mathbf{41.3} \pm \mathbf{5.4}$ | $34.2 \pm 2.6$ | $\mathbf{1.0} \pm \mathbf{0.0}$ | $1.1 \pm 0.0$ |
| | L2 | $\mathbf{61.3} \pm \mathbf{6.2}$ | $57.4 \pm 6.7$ | $\mathbf{1.0} \pm \mathbf{0.0}$ | $1.1 \pm 0.0$ |
| | PNN | $96.9 \pm 0.1$ | $\mathbf{99.4} \pm \mathbf{0.8}$ | $\mathbf{9.9} \pm \mathbf{0.0}$ | $10.6 \pm 0.0$ |
| | SC1 | $47.0 \pm 5.9$ | $\mathbf{32.3} \pm \mathbf{5.1}$ | $\mathbf{1.0} \pm \mathbf{0.0}$ | $1.1 \pm 0.0$ |
| | SCN | $93.2 \pm 2.8$ | $\mathbf{98.0} \pm \mathbf{1.1}$ | $\mathbf{4.0} \pm \mathbf{0.0}$ | $4.3 \pm 0.0$ |
| AntMaze-1 | EWC | $11.0 \pm 5.9$ | $\mathbf{18.2} \pm \mathbf{3.1}$ | $\mathbf{0.9} \pm \mathbf{0.0}$ | $1.0 \pm 0.0$ |
| | FT1 | $9.2 \pm 2.5$ | $\mathbf{18.3} \pm \mathbf{1.6}$ | $\mathbf{0.9} \pm \mathbf{0.0}$ | $1.0 \pm 0.0$ |
| | FTN | $54.0 \pm 3.1$ | $\mathbf{71.1} \pm \mathbf{5.1}$ | $\mathbf{3.7} \pm \mathbf{0.0}$ | $4.0 \pm 0.0$ |
| | FZ | $19.2 \pm 2.5$ | $\mathbf{24.3} \pm \mathbf{0.9}$ | $\mathbf{0.9} \pm \mathbf{0.0}$ | $1.0 \pm 0.0$ |
| | L2 | $4.6 \pm 2.8$ | $\mathbf{12.3} \pm \mathbf{3.0}$ | $\mathbf{0.9} \pm \mathbf{0.0}$ | $1.0 \pm 0.0$ |
| | PNN | $60.8 \pm 7.4$ | $\mathbf{79.0} \pm \mathbf{3.9}$ | $\mathbf{9.2} \pm \mathbf{0.0}$ | $10.0 \pm 0.0$ |
| | SC1 | $11.3 \pm 2.3$ | $\mathbf{18.0} \pm \mathbf{1.7}$ | $\mathbf{0.9} \pm \mathbf{0.0}$ | $1.0 \pm 0.0$ |
| | SCN | $54.0 \pm 5.0$ | $\mathbf{70.8} \pm \mathbf{1.9}$ | $\mathbf{3.7} \pm \mathbf{0.0}$ | $4.0 \pm 0.0$ |

Table 6: **Performances of BC and HBC on each of the baseline methods (avg. on 3 seeds).** HBC consistently outperforms BC on PER across nearly all CRL methods, with significant gains in more sophisticated approaches such as PNN. Notably, HBC shows superior performance even for challenging methods like EWC and L2, while being only less than 10% more expensive in terms of memory usage. The only exceptions are a few naive and underperforming methods, where the gap is small. This demonstrates HBC as a more effective approach for CRL.

Although HBC introduces a small increase in memory usage (MEM), typically less than $10\%$, this trade-off is minimal compared to the significant performance gains. Even for simpler methods like EWC and L2, HBC demonstrates better PER scores, indicating enhanced retention of previously learned tasks and better adaptation to new ones, which is a key requirement for continual reinforcement learning (CRL).

In both task streams, particularly in more complex settings such as AntMaze-1, HBC manages to reduce catastrophic forgetting and outperform BC consistently. This analysis confirms that HBC offers substantial improvements for CRL across all tested baselines, making it a strong candidate for scaling up to more challenging and dynamic environments.

## D.3 Hierarchical Goal-Conditioned CRL Benchmark

| Task Stream | CRL Method | PER ↑ | BWT ↑ | FWT ↑ | MEM ↓ |
|---|---|---|---|---|---|
| **PointMaze-1** | EWC | 55.1 ± 2.9 | -43.5 ± 3.0 | 0.6 ± 2.3 | 1.0 ± 0.0 |
| | FT1 | 50.0 ± 2.8 | -49.1 ± 3.6 | 1.1 ± 1.9 | 1.0 ± 0.0 |
| | FTN | 99.1 ± 0.8 | 0.0 ± 0.0 | 1.1 ± 1.9 | 4.0 ± 0.0 |
| | FZ | 34.2 ± 2.6 | 0.0 ± 0.0 | -63.8 ± 1.6 | 1.0 ± 0.0 |
| | L2 | 57.4 ± 6.7 | -39.3 ± 6.5 | -1.3 ± 0.2 | 1.0 ± 0.0 |
| | PNN | 99.4 ± 0.8 | 0.0 ± 0.0 | 1.4 ± 1.5 | 9.9 ± 0.0 |
| | SC1 | 32.3 ± 5.1 | -65.7 ± 5.8 | 0.0 ± 0.0 | 1.0 ± 0.0 |
| | SCN | 98.0 ± 1.1 | 0.0 ± 0.0 | 0.0 ± 0.0 | 4.0 ± 0.0 |
| | HiSPO (ours) | 98.0 ± 0.4 | 0.0 ± 0.0 | 0.0 ± 0.0 | 2.3 ± 0.0 |
| **PointMaze-2** | EWC | 59.1 ± 3.3 | -40.5 ± 3.5 | -0.4 ± 0.7 | 1.0 ± 0.0 |
| | FT1 | 56.1 ± 4.2 | -43.5 ± 4.6 | -0.4 ± 0.7 | 1.0 ± 0.0 |
| | FTN | 99.6 ± 0.7 | 0.0 ± 0.0 | -0.4 ± 0.7 | 4.0 ± 0.0 |
| | FZ | 32.3 ± 2.8 | 0.0 ± 0.0 | -67.7 ± 2.8 | 1.0 ± 0.0 |
| | L2 | 55.2 ± 3.4 | -43.2 ± 4.9 | -1.6 ± 1.5 | 1.0 ± 0.0 |
| | PNN | 99.5 ± 0.9 | 0.0 ± 0.0 | -0.5 ± 0.9 | 9.9 ± 0.0 |
| | SC1 | 55.5 ± 2.5 | -44.5 ± 2.5 | 0.0 ± 0.0 | 1.0 ± 0.0 |
| | SCN | 100.0 ± 0.0 | 0.0 ± 0.0 | 0.0 ± 0.0 | 4.0 ± 0.0 |
| | HiSPO (ours) | 99.8 ± 0.4 | 1.6 ± 2.7 | -1.8 ± 2.5 | 1.9 ± 0.1 |

Table 7: **CRL Benchmark for Hierarchical Policies on PointMaze Streams (on 3 seeds).**

| Task Stream | CRL Method | PER ↑ | BWT ↑ | FWT ↑ | MEM ↓ |
|---|---|---|---|---|---|
| **AntMaze-1** | EWC | 18.2 ± 3.1 | 0.0 ± 0.0 | -1.9 ± 0.6 | 1.0 ± 0.0 |
| | FT1 | 18.3 ± 1.6 | -52.8 ± 3.6 | -3.4 ± 1.1 | 1.0 ± 0.0 |
| | FTN | 71.1 ± 5.1 | 0.0 ± 0.0 | -3.4 ± 1.9 | 4.0 ± 0.0 |
| | FZ | 24.3 ± 0.9 | 0.0 ± 0.0 | -50.2 ± 1.6 | 1.0 ± 0.0 |
| | L2 | 12.3 ± 3.0 | 0.0 ± 0.0 | -10.8 ± 0.2 | 1.0 ± 0.0 |
| | SC1 | 18.0 ± 1.7 | -56.5 ± 4.0 | 0.0 ± 0.0 | 0.0 ± 0.0 |
| | SCN | 70.8 ± 1.9 | 0.0 ± 0.0 | 0.0 ± 0.0 | 4.0 ± 0.0 |
| | PNN | 79.0 ± 3.9 | 0.0 ± 0.0 | 4.5 ± 1.4 | 10.0 ± 0.0 |
| | HiSPO (ours) | 74.1 ± 3.2 | 0.0 ± 0.0 | -0.4 ± 0.0 | 2.8 ± 0.0 |
| **AntMaze-2** | EWC | 42.5 ± 5.7 | 0.0 ± 0.0 | 10.3 ± 8.1 | 1.0 ± 0.0 |
| | FT1 | 44.5 ± 6.6 | -41.7 ± 5.8 | 14.4 ± 6.1 | 1.0 ± 0.0 |
| | FTN | 72.8 ± 5.3 | 0.0 ± 0.0 | 1.1 ± 7.6 | 4.0 ± 0.0 |
| | FZ | 24.1 ± 1.6 | 0.0 ± 0.0 | -55.1 ± 12.8 | 1.0 ± 0.0 |
| | L2 | 38.3 ± 6.0 | 0.0 ± 0.0 | 2.8 ± 5.7 | 1.0 ± 0.0 |
| | SC1 | 30.3 ± 2.0 | -41.4 ± 3.2 | 0.0 ± 0.0 | 0.0 ± 0.0 |
| | SCN | 71.7 ± 3.5 | 0.0 ± 0.0 | 0.0 ± 0.0 | 4.0 ± 0.0 |
| | PNN | 85.5 ± 2.4 | 0.0 ± 0.0 | 13.8 ± 2.5 | 10.0 ± 0.0 |
| | HiSPO (ours) | 76.5 ± 3.0 | 0.0 ± 0.0 | 4.8 ± 2.8 | 4.0 ± 0.0 |

Table 8: **CRL Benchmark for Hierarchical Policies on AntMaze Streams (on 3 seeds).**

| Task Stream | CRL Method | PER ↑ | BWT ↑ | FWT ↑ | MEM ↓ |
|---|---|---|---|---|---|
| **VideoGame-1** | FT1 | 59.5 ± 9.8 | -28.6 ± 8.6 | 3.6 ± 7.3 | 1.0 ± 0.0 |
| | FTN | 87.7 ± 2.6 | 0.0 ± 0.0 | 4.0 ± 7.1 | 4.0 ± 0.0 |
| | FZ | 54.7 ± 2.7 | 0.0 ± 0.0 | -29.2 ± 9.4 | 1.0 ± 0.0 |
| | PNN | 85.8 ± 2.1 | 0.0 ± 0.0 | 1.4 ± 8.5 | 10.0 ± 0.0 |
| | SC1 | 53.6 ± 4.2 | -30.9 ± 5.7 | 0.0 ± 0.0 | 1.0 ± 0.0 |
| | SCN | 82.8 ± 7.2 | 0.0 ± 0.0 | 0.0 ± 0.0 | 4.0 ± 0.0 |
| | EWC | 65.1 ± 4.0 | -22.8 ± 5.5 | 3.4 ± 7.8 | 1.0 ± 0.0 |
| | L2 | 64.6 ± 5.6 | -15.2 ± 6.2 | -4.7 ± 9.5 | 1.0 ± 0.0 |
| | HiSPO | 87.8 ± 3.5 | 0.0 ± 0.0 | 3.3 ± 9.2 | 2.6 ± 0.0 |
| **VideoGame-2** | FT1 | 63.7 ± 6.9 | -26.7 ± 9.3 | 6.2 ± 1.1 | 1.0 ± 0.0 |
| | FTN | 90.5 ± 2.5 | 0.0 ± 0.0 | 6.3 ± 1.7 | 1.7 ± 0.0 |
| | FZ | 45.8 ± 6.1 | 0.0 ± 0.0 | -37.0 ± 2.7 | 2.7 ± 0.0 |
| | PNN | 86.7 ± 1.4 | 0.0 ± 0.0 | 2.1 ± 1.0 | 10.0 ± 0.0 |
| | SC1 | 64.0 ± 2.6 | -20.3 ± 5.3 | 0.0 ± 0.0 | 1.0 ± 0.0 |
| | SCN | 84.7 ± 4.0 | 0.0 ± 0.0 | 0.0 ± 0.0 | 4.0 ± 0.0 |
| | EWC | 62.2 ± 1.4 | -27.8 ± 3.1 | 5.8 ± 1.9 | 1.9 ± 0.0 |
| | L2 | 66.5 ± 4.3 | -12.5 ± 5.1 | -5.2 ± 2.7 | 2.7 ± 0.0 |
| | HiSPO | 90.2 ± 5.4 | 0.0 ± 0.0 | 5.9 ± 3.3 | 3.3 ± 0.0 |

Table 9: **CRL Benchmark for Hierarchical Policies on Video Game Streams (on 3 seeds).**

## E  ADDITIONAL EXPERIMENTAL DETAILS

### E.1  ANCHOR WEIGHT SAMPLING

Efficient sampling of anchor weights is essential for exploring a policy subspace. We employ a Dirichlet distributions (Ng et al., 2011) in order to uniformly sample weights within a simplex. Using a symmetric Dirichlet distribution with equal concentration parameters facilitates unbiased exploration across the simplex. To enhance sampling efficiency, we implement the stick-breaking process (Paisley, 2010), which accelerates the generation of anchor weights.

Figure 10 illustrates the effectiveness of our sampling method. Subfigure (a) shows the sampling time in an $N$-dimensional simplex, demonstrating the scalability of our approach. Subfigure (b) displays the coverage of the simplex with three anchors, confirming uniform exploration.

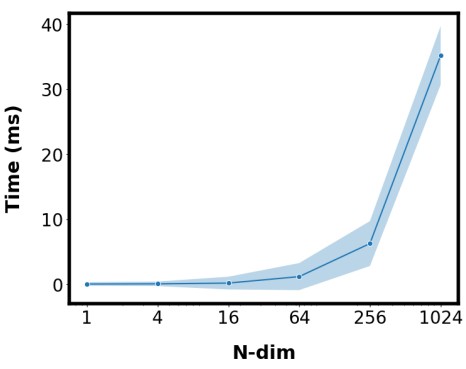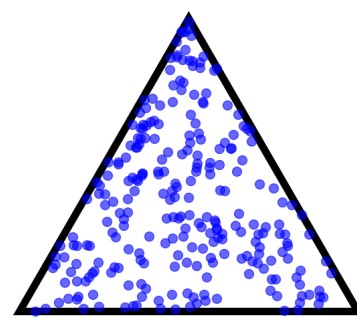

(a) Time to Sample Anchors in the N-dim Simplex (Batch Size = 256, 1000 Reps).

(b) Illustration of the coverage of a 3-Anchor Simplex via Stick Breaking.

Figure 10: Anchor Weight Sampling Illustrations.

While alternative methods, such as gradient-based optimization over the simplex, could be considered, they introduce higher computational costs and risks of converging to local minima. Our Dirichlet-based sampling method ensures extensive coverage of the weight space with manageable computational overhead, making it well-suited for our offline evaluation framework.

### E.2  COMPUTATIONAL COMPLEXITY AND EFFICIENCY

Evaluating the computational efficiency of our HiSPO framework is essential to demonstrate its practicality in continual offline goal-conditioned reinforcement learning. Our experiments across three sets of task streams — PointMaze, AntMaze, and Godot — show that HiSPO introduces minimal additional complexity compared to baseline methods, with the main overhead coming from the evaluation of sampled anchor weights within the policy subspace.

In contrast, methods like CSP and PNN incur higher computational costs. CSP slows computation by requiring the learning of a value function, while PNN's ever-growing architecture requires learning connectors to previous layers outputs during both training and inference. These factors result in significant overhead, especially in high-dimensional environments such as Godot.

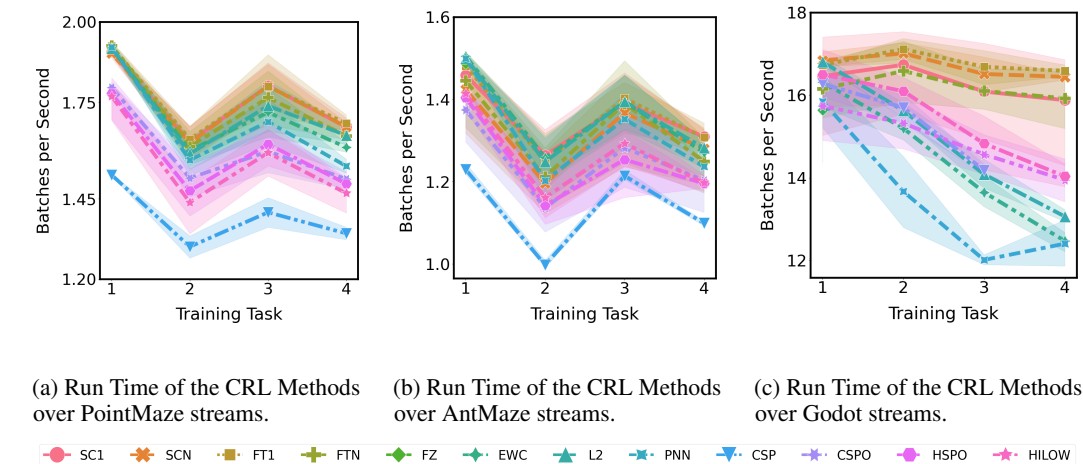

(a) Run Time of the CRL Methods over PointMaze streams.

(b) Run Time of the CRL Methods over AntMaze streams.

(c) Run Time of the CRL Methods over Godot streams.

Figure 11: Anchor Weight Sampling Illustrations.

Figure 11 illustrates the run-time performance of HiSPO compared to baseline methods across the task streams, which maintains competitive run-time efficiency.

