# OpenReview forum: "Hierarchical Subspaces of Policies for Continual Offline Reinforcement Learning"
_ICLR.cc/2025/Workshop/MCDC — MCDC @ ICLR 2025_

### Official Review · Reviewer_Fe4j · 2025-02-17

**Rating:** 3
**Confidence:** 4
**Fit:** 3

**Summary:**

This submission focuses on using policy subspaces for offline continual reinforcement learning. The setting is sequentially observed tasks, and keeping data from old tasks is not permitted. The proposed algorithm, HISPO, is similar to CSP with the exception that it is a two-level policy; one level focuses on sub-goal selection and the other focuses on reaching sub-goals (or goals more generally). HiSPO is evaluated in two sets of 2D gymnasium grid-world and 3D video game maze environments.

**Reason For Giving A Higher Score:**

If the results were better (and included more extensive baselines) or the technical novelty of HiSPO was more significant, I would recommend acceptance.

**Reason For Giving A Lower Score:**

As I have covered in the weaknesses section, this submission needs more work on all frontiers; writing/motivation, technical novelty and evaluation.

**Strengths And Weaknesses:**

---

### Strengths

1. The paper includes extended evaluations.
2. The paper is mostly well-written, with the exception of §4.
3. The paper includes ample explanations and examples from environments, baselines and training details in the Appendix.

---

### Weaknesses

While I like the overall direction of this research, I believe it needs more work before publishing. There are missing baselines, the experimental gains are marginal, and the overall motivation for the paper needs more work.

1. The set of baselines is incomplete. The current baselines mostly include strawmans (SC1, SCN, FT1, FTN, FZ, L2) and only two CRL baselines, EWC and PNN. Given the problem setting, the most natural baseline to consider are Progress and Compress [1] (different work than PNN) and Orthogonal Gradient Descent [2].
2. Improvements over baselines are marginal. The FTN strawman has a slight memory footprint growth compared to HiSPO with similar performance (indistinguishable from confidence intervals based on Tables 7-9). Considering how simple this strawman is, this weakens the empirical edge of HiSPO. Positive empirical results are not necessary for an interesting submission and technical novelty can often overcome negative experimental results; But HiSPO is mostly similar to CSP, and the technical novelty of this work cannot overcome the issues with evaluations.
3. The paper mentions in line 52-53 that CSP is untested in offline CRL and may face new challenges. It would be expected to motivate the investigation with what these new challenges might be. Considering how empirically there does not seem to be much improvement with CSP-like techniques in §5, motivating why the investigation is necessary becomes even more important.
4. I'm somewhat confused on what §5.5.2 and §5.5.3 teach us.\
§5.5.2 suggests that were we to use LoRA, memory footprint would be smaller; I feel this is trivial, and authors should mention why they believe that to not be the case.\
§5.5.3 suggests a PAC learnability criteria for zero-shot subspace evaluation. However, the PAC criteria focuses on some distance function in action space, and **proximity in action space does not imply proximity in returns** (unless actions fully match, which means the environments were identical). This is due to the stateful nature of RL; minor deviations in actions in a trajectory can lead to noticeably different outcomes, i.e., the butterfly effect. Thus, this PAC learnability criteria, which is mentioned as a suggestion at this submission, does not prompt good discussion.
5. A link is mentioned in abstract with no explanation. This usually suggests the link includes source codes or evaluation runs or examples of policy rollouts. However, this link is is empty at time of review, and only includes the abstract. I will give the benefit of the doubt to authors and assume they ran out of time to update the code repo or website, but in general, including a link without any substance is considered misleading, if not malicious.

---

### Minor issues:

1. I find the motivation for the problem setting itself to be lacking, though this does not impact my decision; I merely mention this here as feedback.\
The problem setting is sequential offline CRL, where an offline dataset of trajectories is observed from each task one by one, and the dataset must be removed before observing the next task. This removal of datasets is the part that makes this problem setting quite niche.\
If the issue is data storage, the memory footprint of keeping datasets can be included in empirical evaluations. Furthermore, one can be selective when storing trajectories from tasks for long-term use and not keep entire datasets.\
If the issue is privacy, perhaps detailed examples with citations would be more convincing.
2. Writing issues: Missing explanation for $\tilde{\theta}_k$ in line 143 or FTN in line 318.

[1] Schwarz, Jonathan, et al. "Progress & compress: A scalable framework for continual learning." International conference on machine learning. PMLR, 2018.

[2] Farajtabar, Mehrdad, et al. "Orthogonal gradient descent for continual learning." International Conference on Artificial Intelligence and Statistics. PMLR, 2020.

**Suggestions:**

I have four suggestions:

1. Extend your set of baselines to include more recent baselines as well as P&C and OGD.
2. Amend how HiSPO works to improve its empirical edge in the experiments.
3. Rewrite some text in the introduction to better motivate this problem setup and why studying CSP in offline CRL is interesting in the first place.
4. Either remove §5.5.2/§5.5.3, or expand on them to include non-trivial discussion.

---

### Official Review · Reviewer_goy4 · 2025-02-26

**Rating:** 6
**Confidence:** 3
**Fit:** 4

**Summary:**

This paper proposed Hierarchical Subspaces of Policies (HiSPO), which essentially applies Continual Subspace of Policies (CSP) into offline continual reinforcement learning for navigation tasks. HiSPO decomposes the policy into two layers, a high-level path-planning policy and a low-level path-following policy. When training the policy for a new task, HiSPO applies the concept of CSP, which leverages policy subspaces to preserve previously acquired skills while flexibly adapts to new tasks. More specifically, HiSPO first trains a new anchor weight with efficient exploration in the existing weight subspace, and then evaluates whether the existing anchor weights can be reused so that the new weight can be pruned. The paper demonstrates the effectiveness of HiSPO through experiments in both classical and complex video game-like navigation environments, showing competitive performance while maintaining scalability and efficient memory usage.

**Reason For Giving A Higher Score:**

None

**Reason For Giving A Lower Score:**

None

**Strengths And Weaknesses:**

Strengths:

- The paper provides a comprehensive comparison with many classical CRL algorithms across different navigation tasks in both classical and video game-like environments. The proposed method demonstrates good performance while maintaining a relatively low memory size.
- The paper preliminarily validates additional improvements, such as the incorporation of LoRA.

Weaknesses:

- The paper is somewhat hard to follow. Section 3 Preliminaries could be expanded to provide clearer definitions and explanations. For example, the definition of $\tilda \theta$ in CRL and the meanings of BWT and FWT are not fully explained.
- Section 5.5.3 lacks experimental support.
- HiSPO performs badly with regard to FWT. This suggests that HiSPO may struggle with generalizing to entirely new tasks or environments.

**Suggestions:**

- Consider improving the cohesion of the writing to make the paper flow more naturally. The authors can provide more detailed descriptions of their methodology and assumptions, especially in the early sections of the paper, so readers unfamiliar with the specific terms can grasp the paper's full meaning.
- Section 5.5.3 needs to be supported with experimental results to strengthen the claims made in that part of the paper.
- The authors may include real-world navigation tasks to further demonstrate the generalizability and efficiency of HiSPO.

---

### Official Review · Reviewer_BYYJ · 2025-03-02

**Rating:** 6
**Confidence:** 3
**Fit:** 4

**Summary:**

This paper presents a new method for continual offline RL, in which "anchor" parameter sets are progressively added as new tasks are seen during training.  The decision for whether new anchors are added or not is based on whether the loss is reduced by updating the subspace with the new anchor.  The algorithm is fairly simple and provides modest improvements over some previous methods, and the tradeoff between performance and memory usage is explored.

notes from reading the paper:
  -Hierarchical subspaces of policies for continual offline-RL.
  -Task where agent must adapt to new tasks while retaining previously acquired skills.
  -Challenge is avoiding forgetting past gathered knowledge.  Needs to be scalable as new tasks are added.
  -One domain where this comes up is in navigation, where topology or kinematics can change.
  -mujoco maze and video game navigation.
  -Focuses on goal-conditioned rl.
  -Continual subspace of policies (Gaya 2023).
  -Learn a simplex weighting over different anchor parameters.
  -Contribution involves growing separate parameter subspaces for a high-level path planning policy and a low-level path-following policy.
  -High level policy predicts sub-goal k steps in the future.
  -Uses hindsight experience replay.
  -Add new anchor if the new subspace lowers loss.
  -replay buffer seems fine to me.  I'm skeptical that this is so bad, although I understand that this is an established issue in this area.

**Reason For Giving A Higher Score:**

The paper is well-written and the method is explained well.  The results are solid, establishing a new trade-off between memory usage and performance on learning from multiple tasks.

**Reason For Giving A Lower Score:**

More analysis could be given, for example showing an example problem showing how different anchors learn diverse subgoals, and how this benefits continual learning.  Additionally the results are relatively modest given the complexity of the method.

**Strengths And Weaknesses:**

Strengths:
  -The paper is well-written and explains both the problem and the method well.

Weaknesses:
  -Performance not much better than SCN baseline, while the SCN baseline seems much simpler.
  -There is not much analysis of what the anchors learn.

**Suggestions:**

Maybe it would be nice to give some analysis of what different anchors learn, to show that the pruning of some new anchors is useful.  Additionally, it would be nice to see that the anchors are actually diverse and reflect different skills.

---

### Official Review · Reviewer_wsXK · 2025-03-04

**Rating:** 6
**Confidence:** 3
**Fit:** 4

**Summary:**

The paper explores the use of Continual Subspace of Policies (CSP) in the context of continual offline goal-conditioned imitation learning. It introduces a subspace for both high-level and low-level policies, aiming to maintain performance while minimizing memory usage.

**Reason For Giving A Higher Score:**

See Strength

**Reason For Giving A Lower Score:**

See Weakness

**Strengths And Weaknesses:**

## Strength
1. The paper is clear and well written.
2. The proposed hierarchical subspace of policy is novel and performs better than other baselines.

## Weakness
1. The paper template requires a maximum of 6 pages, but the paper submitted is 8 pages.
2. When encountering a new task, it is necessary to first train new parameters before deciding whether to retain them. If they are not retained, the new training process would be completely wasted.

**Suggestions:**

Revise the paper to meet the required length of 6 pages.

---

### Decision · Program_Chairs · 2025-03-06

**Decision:**

Accept

**Comment:**

This work proposes a hierarchical continual learning method for RL, which has some relevance to this workshop. 3 over 4 reviewers recommend acceptance, therefore we are accepting this work to the workshop. However in the spirit of peer reviewing improving the state of research, we strongly recommend the authors to take notes of the suggestions from reviewer Fe4j, they are directly actionable and would strengthen this work.